# G-CSF/NAMPT signaling drives neutrophil dysfunction and enhances bacterial infection susceptibility in cancer patients

Ekaterina Pylaeva [1,2], Lea Tollrian[1], Jana Riedesel[1], Olga Shevchuk [3], Ilona Thiel[1], Irem Ozel [1], Nastassia Kabankova [1], Hannah Voß[3], Bente Siebels[4], Hartmut Schlüter [4], Corinna Haist[5], Helmut Hanenberg[5,6], Stefan Mattheis[1], Cornelius Kürten[1], Markus Sperandio [7], Jan Kehrmann[8], Daniel Robert Engel[3], Stephan Lang[1,2] & Jadwiga Jablonska [1,2] ✉

Despite advancements in cancer therapies, bacterial complications remain a major challenge, delaying treatment and worsening outcomes. While immunosuppressive therapies and prolonged hospitalizations contribute, they do not fully explain the elevated infection risk in cancer patients. Here we show that tumors producing high levels of granulocyte colony-stimulating factor (G-CSF) promote the persistence of Gram-negative pathogens in head and neck squamous cell carcinoma due to neutrophil reprogramming. Mechanistically, we identify tumor-driven activation of the G-CSF / nicotinamide phosphoribosyltransferase (NAMPT) signaling axis in neutrophil progenitors, resulting in impaired antibacterial functions, such as phagocytosis and neutrophil extracellular traps formation, and development of tissue-damaging neutrophil subsets. This disrupts lung tissue integrity and facilitates bacterial persistence. Importantly, targeting the G-CSF/NAMPT pathway prevents the generation of dysfunctional neutrophils and improves bacterial clearance in vivo. Our findings reveal tumor-induced, NAMPT-dependent neutrophil reprogramming as a central driver of compromised antimicrobial defenses in cancer. Therapeutic strategies aimed at modulating G-CSF/NAMPT signaling could enhance infection control and survival for cancer patients.

In patients with cancer, infectious complications significantly impact clinical outcomes, delaying treatment and increasing morbidity and mortality. Among these, infections caused by *Pseudomonas aeruginosa* represent a particularly serious threat, with cancer patients experiencing a 50-fold higher risk of bacteremia compared to the general population.[1] Alongside other Gram-negative pathogens such as *Klebsiella pneumoniae*, *Escherichia coli*, and *Haemophilus influenzae*, *P. aeruginosa* remains a high-priority target in the 2024 Bacterial Priority Pathogens List due to its high rates of antibiotic resistance and global prevalence, especially in healthcare settings. The World Health

[1]Department of Otorhinolaryngology, Head and Neck Surgery, University Hospital Essen, University of Duisburg-Essen, Essen, Germany. [2]German Cancer Consortium (DKTK) partner site Düsseldorf/Essen, Essen, Germany. [3]Department of Immunodynamics, Institute for Experimental Immunology and Imaging, University Hospital Essen, University of Duisburg-Essen, Essen, Germany. [4]Section Mass Spectrometry and Proteomics, Center for Diagnostics, University Medical Center Hamburg-Eppendorf (UKE), Hamburg, Germany. [5]Department of Otorhinolaryngology and Head/Neck Surgery, Heinrich Heine University, Düsseldorf, Germany. [6]Department of Pediatrics III, University Hospital Essen, University of Duisburg-Essen, Essen, Germany. [7]Institute of Cardiovascular Physiology and Pathophysiology, Biomedical Center, Ludwig Maximilian University of Munich, Munich, Germany. [8]Institute of Medical Microbiology, University Hospital Essen, University of Duisburg-Essen, Essen, Germany. ✉e-mail: Jadwiga.Jablonska@uk-essen.de

Organization underscores the urgent need for innovative prevention and control measures, as well as alternative therapeutic strategies, to combat these pathogens effectively.[2]

While traditional risk factors, including prolonged hospitalization and immunosuppressive treatments, contribute to the increased infection burden in cancer patients, they do not fully explain the vulnerability observed even in untreated cases. Emerging evidence suggests that tumors themselves may drive an intrinsic reprogramming of immune functionality, significantly impairing the body's ability to resist bacterial infections.[3,4] This raises critical questions about the mechanisms of immune dysfunction in cancer and their implications for both infection control and therapeutic intervention.

Neutrophils, central to antibacterial immunity, have historically been viewed as short-lived innate effector cells with limited adaptability.[5,6] However, recent discoveries in the field of trained immunity challenge this perspective, demonstrating that innate immune cells, including neutrophils, can acquire a form of memory through epigenetic modulation following specific stimuli.[7] While trained immunity enhances responses to subsequent infections[8] or even malignancies,[9] the tumor microenvironment appears to induce a distinct phenomenon: immune reprogramming. Tumor-derived factors drive this reprogramming, potentially resulting in maladaptive immune responses that compromise the host's ability to clear bacterial infections.[10,11] Despite its clinical significance, the long-term impact of tumor-driven innate immune reprogramming on neutrophil function remains poorly understood.

Here, we show that tumor-derived granulocyte colony-stimulating factor (G-CSF) reprograms neutrophils through activation of the NAMPT-NAD signaling axis, leading to cytoskeletal alterations, impaired antibacterial responses, and accumulation of tissue-toxic neutrophil subsets. We demonstrate that this neutrophil dysfunction increases susceptibility to Gram-negative bacterial infections in cancer hosts. Moreover, targeting G-CSF-NAMPT signaling restores neutrophil antibacterial function and improves infection control. These findings identify NAMPT-dependent neutrophil reprogramming as a central driver of compromised antimicrobial defense in cancer, providing a rationale for therapeutic strategies targeting this pathway.

## Results

### Increased susceptibility of cancer patients to bacterial infections is associated with high G-CSF release from tumor tissue

Cancer patients suffer from recurring bacterial infections. To assess the abundance of clinically relevant Gram-negative pathogens in such patients (*P. aeruginosa, E. coli, K. pneumonia, H. influenza*),[12,13] we performed microbiological analysis of oral rinse of clinically asymptomatic HNSCC patients and compared it to healthy individuals (Table 1). Of note, the frequency of Gram-negative pathogens was significantly higher in such patients (Fig. 1a). None of the recorded clinical parameters (age, sex, smoking status, HPV) was significantly associated with the presence of Gram-negative pathogens in the oral cavity (Table 2).

Moreover, in a 2-year prospective study, we demonstrated that the presence of Gram-negative pathogens in the oral cavity indicated the poor 2-year outcome (Fig. 1b), associated with significantly increased risk of bacterial complications (Fig. 1c, clinical characteristics in Table 3).

Given that bacterial infections are mainly controlled by neutrophil granulocytes and their antibacterial functions are modulated by G-CSF, we assessed blood and tumor levels of G-CSF in cancer patients, and observed a significant increase in tumor tissue (Fig. 1d). Moreover, G-CSF levels were also increased in the oral cavity of patients, compared to healthy individuals (Fig. 1e).

Next, we evaluated whether G-CSF levels in the oral cavity correlate with patient susceptibility to bacterial infections. Using a received operating characteristic (ROC) analysis, we determined the cutoff

**Table 1 | Clinicopathological characteristics of the study participants**

|  |  | Healthy $n = 28$ | HNC $n = 45$ |
|---|---|---|---|
| Male, % |  | 52% | 78% |
| Mean age, years |  | 61 | 64 |
| Active smokers, % |  | 35% | 65% |
| Mean pack-years |  | 14.8 | 35.5 |
| HPV positive, % |  | - | 38% |
| UICC stage | I | - | 13 |
|  | II | - | 12 |
|  | III | - | 3 |
|  | IV | - | 17 |
| Tumor localization | Oral, oropharynx | - | 28 |
|  | Hypopharynx | - | 6 |
|  | Larynx | - | 10 |
|  | Other | - | 1 |

Baseline demographic, clinical, and tumor-related characteristics of healthy controls ($n = 28$) and patients with head and neck carcinoma (HNC; $n = 45$). Values are presented as percentages or means, as indicated. Smoking history is expressed as mean pack-years. HPV status was determined in tumor tissue, and the UICC stage was reported at diagnosis. Tumor localization refers to the primary site at presentation.
*HNC* head and neck carcinoma, *HPV* human papillomavirus, *UICC* Union for International Cancer Control.

G-CSF value in oral rinse as 392 pg/ml, with the highest sensitivity (75%) and specificity (82.3%), indicating the presence of Gram-negative pathogens in the oral cavity (Fig. 1f). Indeed, we observed higher frequencies of Gram-negative pathogens in patients with high G-CSF levels (Fig. 1g), with positive likelihood ratio 4.25 (95% CI 1.34–11.96) (Fig. 1h). Of note, in the general population (both, healthy and HNSCC) increased G-CSF levels in oral rinse (400 pg/ml or higher) predict the presence of Gram-negative pathogens with a specificity of 91% and a positive likelihood ratio of 8.4 (95% CI 2.64–26.71) (Fig. S1).

### G-CSF, chronically released by the tumor tissue, promotes bacterial infections

Short-term exposure to G-CSF is reported to have antibacterial and neutrophil-stimulating properties.[14] In agreement, treatment of isolated human blood neutrophils with G-CSF results in elevated survival and antibacterial responses of these cells (elevated NETs production) with the lack of effect on ROS production and phagocytosis (Fig. S2a–c).

However, tumor presence is associated more with long-lasting G-CSF release, leading us to hypothesize that it may have a detrimental effect on neutrophil properties. To test this, we modified a murine oropharyngeal carcinoma (MOPC[15]) cell line to overexpress G-CSF (GMOPC) at levels comparable to those measured in patients' tumors (Fig. 2a).

To evaluate whether chronically elevated G-CSF impacts susceptibility to lower respiratory tract bacterial infections, we infected tumor-bearing mice (MOPC versus GMOPC) intratracheally (*i.t.*) with *P. aeruginosa*, and followed the course of infection. Indeed, similarly to patients, elevated chronic G-CSF availability (GMOPC) increased the susceptibility of tumor-bearing animals to infection (Fig. 2b). This was accompanied by the elevated lung tissue damage and loss of aerated area (Fig. 2c, d) in these mice. Moreover, higher levels of TNFα as a marker of tissue damage (Fig. 2e) and augmented severity of the disease (Fig. 2f) were observed.

### Chronic exposure to tumor-derived G-CSF impairs neutrophil antibacterial functions and facilitates their tissue toxicity

G-CSF, chronically released by the tumor tissue, stimulates granulopoiesis and neutrophil release from the bone marrow, resulting in elevated neutrophil accumulation in different compartments, such as tumor, lung (Fig. 2g, gating strategy Fig. S3), as well as in bone marrow

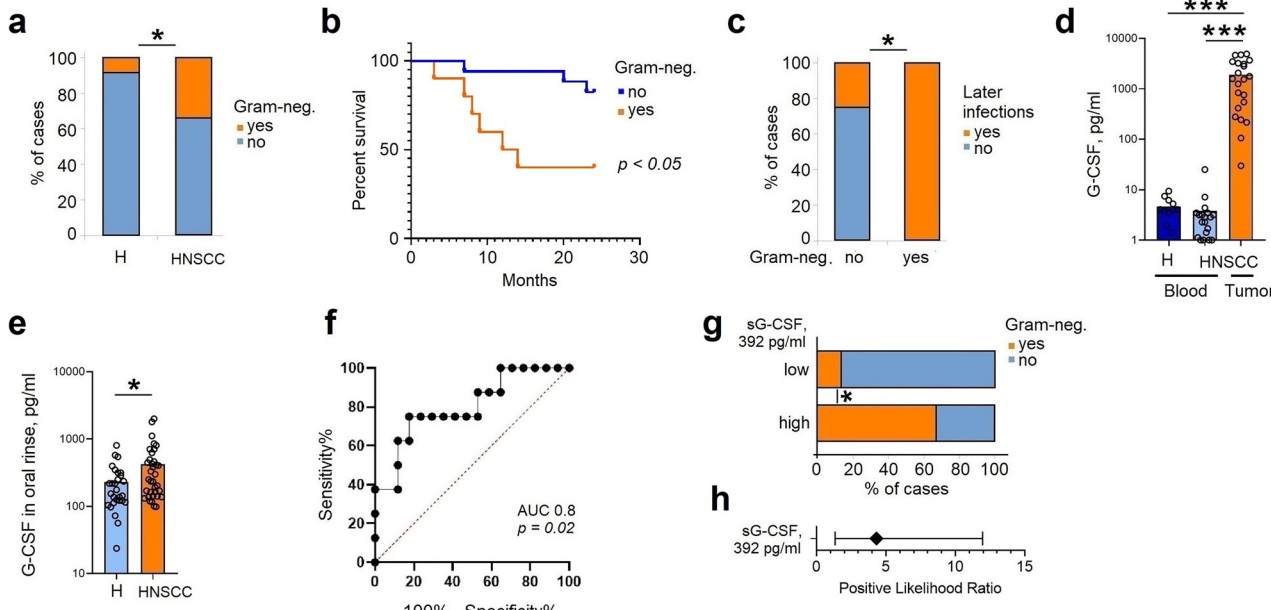

**Fig. 1 | High G-CSF release by tumors is associated with persistence of Gram-negative pathogens in HNSCC. a–h** Oral rinse samples (50 ml sterile saline) were collected from healthy volunteers (*n* = 28) and patients with head and neck squamous cell carcinoma (HNSCC; *n* = 45) immediately after awakening, prior to oral hygiene or food and drink intake. Frequencies of Gram-negative pathogens (Escherichia coli, Klebsiella pneumoniae, Haemophilus influenzae, Pseudomonas aeruginosa) were determined by culture on universal and selective agar media in *n* = 23 healthy and *n* = 32 participating patients. Granulocyte colony-stimulating factor (G-CSF) concentrations were measured with ELISA. Clinical follow-up was conducted over 2 years. **a** Frequency of clinically significant Gram-negative pathogens is higher in untreated HNSCC patients than in healthy controls. **b** In HNSCC, Gram-negative pathogen carriage at baseline correlates with poorer 2-year survival. **c** In HNSCC, Gram-negative pathogen carriage at baseline is associated with increased risk of bacterial infections over a 2-year follow-up. **d** HNSCC tumors express high levels of G-CSF as estimated in tumor-conditioned medium, in comparison to plasma. **e** G-CSF concentrations in oral rinse are elevated in HNSCC patients (*n* = 45) compared to healthy (*n* = 28). **f** Receiver operating characteristic (ROC) analysis of G-CSF levels in oral rinse as a predictor of Gram-negative pathogen presence in HNSCC. **g** G-CSF levels above 392 pg ml$^{-1}$ in oral rinse predict Gram-negative pathogen detection in HNSCC. **h** | Patients with G-CSF concentrations >392 pg ml$^{-1}$ have a 4.25-fold increased risk of Gram-negative pathogen detection, error bars represent 95% CI. H healthy controls, HNSCC head and neck squamous cell carcinoma, G-CSF granulocyte colony-stimulating factor, AUC area under the curve. Data are presented as means ± individual data points. Statistical tests: $\chi^2$ (**a**, **c**, **g**); Kruskal–Wallis for multiple-group comparisons with Bonferroni correction, and two-sided Mann–Whitney for two-group comparisons (**d**, **e**); comparison of Kaplan–Meier survival curves by Log-rank (Mantel–Cox) test (**b**). *$p$ < 0.05, **$p$ < 0.01, ***$p$ < 0.001. Source data and exact $p$-values are provided as a Source Data file.

## Table 2 | Association of the presence of Gram-negative pathogens in the oral cavity with HNSCC risk factors

| Parameter | No Gram-negative Pathogens | Yes Gram-negative Pathogens | *p*-value |
|---|---|---|---|
| Mean Age (years) | 65 | 66.3 | 0.87 |
| Male Sex (%) | 87.50% | 83.30% | 0.73 |
| Current Smokers (%) | 73.70% | 50.00% | 0.2 |
| Mean Pack-Years | 43.3 | 29.9 | 0.25 |
| HPV Positive (%) | 44.40% | 44.40% | 1 |

Two-sided Mann–Whitney U-test and χ2 tests were used for comparison of two independent samples. None of these parameters—including sex and current smoking status—were significantly associated with the presence of Gram-negative pathogens in the oral cavity.

and blood (Fig. S4). A similar phenomenon is observed in HNSCC patients, where a higher accumulation of neutrophils in blood, tumor, or oral rinse positively correlates with elevated G-CSF levels (Fig. S5). Notably, the predominant neutrophil subpopulations associated with elevated G-CSF levels were CD62L$^{low}$ CD11b$^{dim}$ senescent subsets (Fig. 2h).

To investigate the molecular mechanism behind possibly dysregulated antibacterial activity of neutrophils chronically exposed to tumor-derived G-CSF, we compared the proteome of lung neutrophils isolated from mice bearing $^G$MOPC (further referred to as "$^G$neutrophils") versus MOPC tumors ("neutrophils"). By quantitative proteomics, we identified 115 differentially expressed proteins (63 upregulated versus 52 downregulated proteins) in $^G$neutrophils (Supplementary Data 1). Unsupervised hierarchical clustering analysis showed a unique proteomic signature of neutrophils under chronic exposure to G-CSF (Fig. 2h, i).

One of the molecules significantly downregulated in $^G$neutrophils is CD11b (integrin αM), which plays a major role in neutrophil antibacterial activity. We validated the robustness of our label-free quantitative proteomic analysis and confirmed the downregulation of CD11b under chronic G-CSF exposure in $^G$neutrophils using flow cytometry (Fig. 2h, Fig. S6).

Next, we performed gene ontology (GO) enrichment analysis of differentially expressed proteins to gain insights into the biological processes (Fig. 2j), cellular compartments (Fig. 2k), and molecular functions (Fig. 2l) that are affected in neutrophils by the long-term G-CSF exposure. Proteins upregulated in such $^G$neutrophils were significantly enriched in the top 5 GO categories linking to neutrophil transcriptional activity, suggesting a possible dysregulation in their maturation, whereas proteins downregulated in $^G$neutrophils were associated with cytoskeleton reorganization (Fig. 2j–l, Supplementary Data 2–7).

### $^G$Neutrophils show impaired phagocytosis and NET formation due to defects in cytoskeleton polymerization

In agreement with the altered phenotype, we observed a significant downregulation of pathways involved in *actin polymerization and*

**Table 3 | Infectious complications in participating patients with head and neck carcinoma (HNC)**

| Group | Mean age (min-max) at admission, years | Mean time interval (min-max), months | Source of infection | Infectious agent | Gram-neg bacteria at admission |
|---|---|---|---|---|---|
| No infection (n = 10) | 62.2 (49–76) | 24.5 (7–36) | - | - | 60% (no), 0% (yes), 40% (NA) |
| Later infection (n = 8) | 54.7 (40–77) | 4 (1–12) | See below | See below | 12.5% (no), 62.5% (yes), 25% (NA) |
| ID19 | | | Tracheal secret, bronchial secret | *E. cloacae* | Yes |
| ID25 | | | Throat | *E. coli* | Yes |
| ID26 | | | Breast | *A. lwoffi* | Yes |
| ID30 | | | Blood (pneumogenic) | *NA* | No |
| ID32 | | | Blood | *B. cereus, S. oralis, E. coli, K. pneumoniae* | Yes |
| ID40 | | | Odontogenic | *NA* | NA |
| ID42 | | | Pneumonia | *E. coli* | NA |
| ID50 | | | Demerskatheter, BAL | *S. epidermidis, Candida* | Yes |

Baseline characteristics, infection timing, source, pathogenic agents, and Gram-negative bacterial status at admission for patients with and without infectious complications during follow-up. Mean age (range) at hospital admission, mean time interval (range) in months from diagnosis to infection, and detailed infection source and causative agents are shown where available. Gram-negative bacterial status at admission is expressed as the percentage of patients without Gram-negative bacteria (no), with Gram-negative bacteria (yes), or with data not available (NA).

*HNC* head and neck carcinoma, *NA* data not available, *BAL* bronchoalveolar lavage, *E. cloacae* Enterobacter cloacae, *E. coli* Escherichia coli, *A. lwoffi* Acinetobacter lwoffii, *B. cereus* Bacillus cereus, *S. oralis* Streptococcus oralis, *K. pneumoniae* Klebsiella pneumoniae, *S. epidermidis* Staphylococcus epidermidis.

*phagocytosis* in [G]neutrophils (Fig. 3a, blue). While total expression of actin was higher, actin-regulatory proteins responsible for actin dynamics (polymerization, depolymerization, and branching), such as Arpc4 (Actin Related Protein 2/3 Complex Subunit 4) or Cap1 (Cyclase Associated Actin Cytoskeleton Regulatory Protein 1), were downregulated (Fig. S7). Apparently, cytoskeleton-dependent functions of [G]neutrophils are impaired due to chronic G-CSF exposure.

To determine where the notable alterations of the cytoskeleton influence antibacterial functions of neutrophils, we challenge these cells with *P. aeruginosa*. We observed significantly reduced phagocytosis (Fig. 3b) and NET formation (Fig. 3c, e, Fig. S7) by [G]neutrophils, but no changes in total actin polarization (Fig. 3d). Treatment of neutrophils with the inhibitor of the Arp2/3 complex CK666 resulted in a significant suppression of phagocytosis, directly confirming that proper actin cytoskeleton reorganization is essential for neutrophil phagocytic function (Fig. S7). These results provide experimental evidence that defects in actin polymerization and cytoskeletal dynamics, as observed in [G]neutrophils, are causally linked to their impaired antibacterial activity.

The expression of other antibacterial proteins, such as Padi4 (Protein-arginine deiminase type-4) and non-oxidative branch of pentose-phosphate pathway (Transaldolase 1 Taldo, Transketolase Tkt) that are involved in NET formation, as well as Cybb (cytochrome b-245, beta chain), involved in phagolysosome pathogen killing, was also decreased upon G-CSF exposure (Fig. S7f–i). This possibly contributes to the impaired bactericidal activity of these cells. Importantly, NET formation by [G]neutrophils in response to *P. aeruginosa* was decreased as compared to neutrophils transiently stimulated by G-CSF (Fig. S2a–c), once again demonstrating different mechanisms involved in chronic versus acute G-CSF stimulation.

**Chronic exposure to G-CSF boosts tissue toxicity of [G]neutrophils during bacterial infection**

[G]Neutrophils show upregulated *Response to the reactive oxygen species* pathway (Fig. 3a, orange). In agreement, we observed an increased expression of proteins involved in ROS production, such as Mpo (Myeloperoxidase, Fig. 3f). As Mpo is a granule protein, high *Mpo* gene expression indicates high de novo synthesis of this protein, rather than decreased degranulation (Fig. 3G). Besides non-mitochondrial ROS, we observe elevated mitochondrial activity in [G]neutrophils (Fig. 3h–i,

Fig. S8a). At the same time, downregulated expression of the components of ROS detoxication machinery, such as Gpx1 (Glutathione peroxidase 1, Fig. S8b), is observed. Such an elevated ROS production, both in steady state and during infection (Fig. 3j–l), can be responsible for the observed lung tissue damage.

To directly evaluate extracellular ROS release, we performed the Amplex Red assay, which specifically detects hydrogen peroxide in the extracellular compartment. Our results confirmed that [G]neutrophils exhibit significantly elevated spontaneous ROS release compared to control neutrophils. Notably, while spontaneous ROS production and release were higher in [G]neutrophils, particularly in CD62L[low] cells, the increase in ROS in response to bacterial stimulation was less pronounced than in normal neutrophils (Fig. S8). This suggests a dysregulated oxidative burst, with higher baseline tissue exposure to ROS but impaired antibacterial response upon challenge.

Another potential cause of lung tissue damage could be an increased release of enzymes involved in matrix remodeling. Proteomic analysis of lung tissue from [G]MOPC-bearing mice identified upregulation of MMP8, MMP9, and MMP25 in lung tissue. These MMPs are implicated in extracellular matrix remodeling and immune regulation. The concurrent upregulation of these three MMPs suggests a broader pattern of matrix-degrading enzyme induction in the context of chronic G-CSF exposure (Fig. S9a). We confirmed significantly elevated levels of MMP9 in lung homogenates of infected animals (Fig. 3m). Consistently, isolated [G]neutrophils release higher amounts of MMP9 in response to *P. aeruginosa* challenge in vitro (Fig. 3n), and also show elevated *Mmp9* gene levels in steady state (Fig. 3o).

Morphological assessment of lung tissue from [G]MOPC-bearing mice revealed several classic features of matrix remodeling, including disrupted distribution of collagen in the lung tissue, thickening of alveolar walls, and increased cellularity within the lung parenchyma, indicating significant collagen degradation (Fig. S9b). In parallel, we observed decreased expression of proteins critical for lung tissue integrity, such as occluding (Fig. S9c), further supporting the presence of tissue injury in this model.

**[G]Neutrophils demonstrate impaired bactericidal activity**

Next, we conducted bacterial killing assays to assess the functional consequences of altered [G]neutrophil properties. Short-term killing (1-h plating) was slightly more effective in [G]neutrophils, indicating that

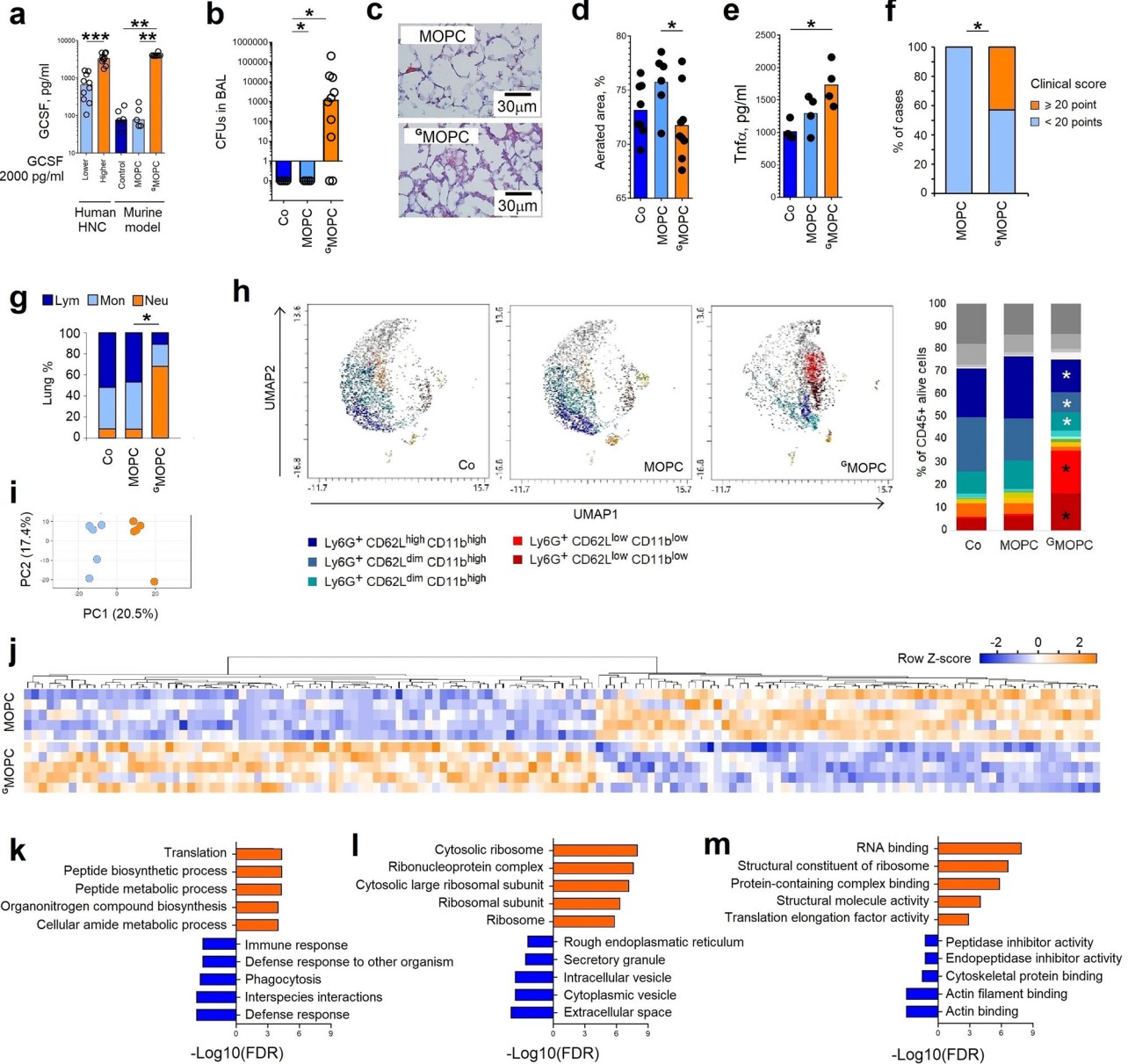

**Fig. 2 | Murine model of HNSCC with high G-CSF production ($^G$MOPC) is characterized by impaired neutrophil bactericidal responses. a** G-CSF concentrations in tumor-condensed medium from HNSCC patients ($n = 20$) and in plasma in murine oropharyngeal carcinoma models with low (MOPC) ($n = 5$ mice) or high ($^G$MOPC) G-CSF production ($n = 8$ mice) in comparison to tumor-free mice ($n = 6$). Samples of tumor-condensed medium from HNSCC patients with low ($n = 10$) and high ($n = 10$) G-CSF content were selected based on median levels in the cohort. **b–f** Mice were injected subcutaneously with PBS (control, Co), MOPC, or $^G$MOPC cells. On day 14, Pseudomonas aeruginosa ($10^6$ CFU) was inoculated intratracheally. After 18 h, animals were sacrificed, and BAL fluid and lung tissue were collected (gating strategy Fig. S3a). **b** Decreased clearance of P. aeruginosa from the lower respiratory tract in $^G$MOPC-bearing mice ($n = 14$) in comparison to MOPC-bearing ($n = 8$) or tumor-free ($n = 4$) mice. **c** Representative image of lung tissue damage (haematoxylin–eosin, scale bar = 30 μm). **d** Reduced aerated lung area in $^G$MOPC-bearing mice ($n = 9$) in comparison to MOPC-bearing ($n = 6$) or tumor-free ($n = 8$) mice. **e** Increased TNFα expression in lungs of $^G$MOPC-bearing mice ($n = 4$) in comparison to MOPC-bearing ($n = 4$) or tumor-free ($n = 4$) mice. **f** Worse general clinical performance in $^G$MOPC-bearing mice ($n = 14$) in comparison to MOPC-bearing ($n = 8$) mice. **g, h** Mice were injected subcutaneously with PBS, MOPC, or $^G$MOPC cells. On day 14, animals were sacrificed, lungs were collected, and live

Ly6G$^+$ neutrophils were isolated. **g** $^G$MOPC-bearing mice ($n = 4$) showed marked neutrophil infiltration in non-infected lungs in comparison to MOPC-bearing ($n = 4$) or tumor-free ($n = 4$) mice. **h** Lungs of $^G$MOPC-bearing mice ($n = 3$) were enriched in CD62L$^{low}$ CD11b$^{dim}$ neutrophil subsets in comparison to lungs of MOPC-bearing ($n = 3$) or tumor-free ($n = 3$) mice. **i–m** Mice were injected subcutaneously with MOPC or $^G$MOPC cells. On day 14, animals were sacrificed, lungs were collected, and live Ly6G$^+$ neutrophils were isolated for proteomic profiling. **i, j** Lung neutrophils from $^G$MOPC-bearing mice (orange) displayed a distinct proteomic signature compared with the MOPC group (blue) ($n = 5$ mice in each group). **k–m** Gene Ontology (GO) enrichment analysis of upregulated (orange) and downregulated (blue) pathways in $^G$MOPC lung neutrophils: **k** biological processes; **l** cellular components; **m** molecular functions. HNSCC head and neck squamous cell carcinoma, Co control tumor-free mice, MOPC mice bearing non-G-CSF-producing tumors, $^G$MOPC mice bearing G-CSF-producing tumors, BAL bronchoalveolar lavage, TNFα tumor necrosis factor-α, CFU colony-forming units. Data are presented as means with individual values (biological replicates) shown. Statistical tests: χ2 (**f**); Kruskal–Wallis for multiple-group comparisons with Bonferroni correction, two-sided Mann–Whitney for two-group comparisons (**a, b, d, e, g**). *$p < 0.05$, **$p < 0.01$, ***$p < 0.001$. Source data and exact $p$-values are provided as a Source Data file.

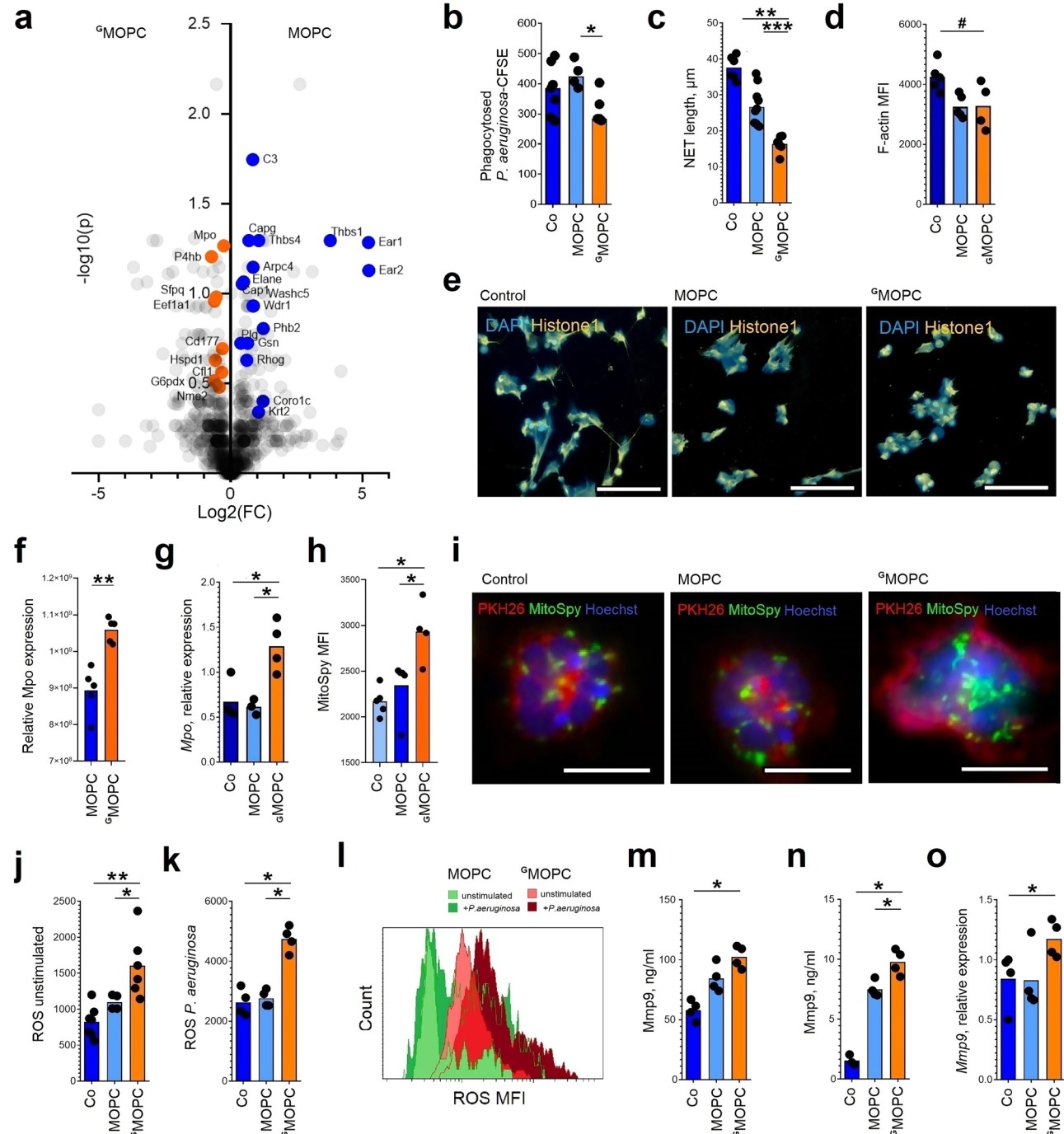

initial antibacterial mechanisms remain active. However, with prolonged exposure (24-h crystal violet assay) (Fig. S10a), there was increased evidence of biofilm formation by bacteria (scale bar 100 μm), suggesting that ᴳneutrophils may fail to sustain effective bacterial clearance over time, possibly due to their functional exhaustion (Fig. S10b).

## ᴳNeutrophils show a distinct aged phenotype

ᴳNeutrophil proteome analyses revealed dysregulation of neutrophil ageing and apoptosis (Fig. 4a). In agreement, we observed significantly decelerated apoptosis of lung ᴳneutrophils (Fig. 4b), accompanied by significantly downregulated Caspase 3 expression (Fig. 4c). At the same time, we observed accumulation of aged Ly6G⁺CD62L^low^CD101^high^CXCR4^high^ polymorphonuclear ᴳneutrophils in lungs (Fig. 4d, Fig. S11) of mice bearing G-CSF-producing tumors. There

was a marked increase in the proportion of aged, CD62L^low^ neutrophils in both bone marrow and blood of ᴳMOPC-bearing mice, mirroring the phenotype seen in lung tissue. The observed alterations in neutrophil phenotype were present in all analyzed compartments—bone marrow, blood, and lung—indicating that the programming initiated by tumor-derived G-CSF is systemic and not restricted to a single tissue (Fig. S4).

Such aged CD62L^low ᴳ^ neutrophils are strongly cytotoxic, with low (degranulation-associated) SSC expression (Fig. 4e), elevated Mmp9 production (Fig. 4f), and enhanced cytotoxicity in vitro (Fig. 4g). In line with this, we observed high abundance of neutrophil granule-associated (azurophilic, specific, gelatinase, secretory vesicles) proteins in lungs of ᴳMOPC-bearing mice (Fig. S9, Fig. S12).

Moreover, these cells distinct exhausted phenotype, with suppressed antibacterial activity, including phagocytosis (Fig. 4h),

**Fig. 3 | Chronic exposure to tumor-derived G-CSF impairs bactericidal activity and increases tissue toxicity of ᴳneutrophils. a** Mice were injected subcutaneously with MOPC or ᴳMOPC cells (*n* = 5 mice in each group). On day 14, animals were sacrificed, lungs were collected, and viable Ly6G⁺ neutrophils (gating strategy Fig. S3a) were isolated for LC-MS/MS proteomic analysis. Volcano plot shows proteins downregulated in ᴳneutrophils (blue; involved in actin polymerization and phagocytosis) and proteins upregulated in ᴳneutrophils (orange; involved in reactive oxygen species responses) according to Signal-to-noise ratio (SNR) > 1. **b**–**l**, **o** Cell properties and functions of isolated viable Ly6G⁺ neutrophils (gating strategy Fig. S3a) were evaluated in vitro in the absence or presence of *Pseudomonas aeruginosa* at MOI 10. **b** Decreased phagocytic capacity of ᴳneutrophils (isolated from *n* = 7 mice) in comparison to lung neutrophils from MOPC-bearing (*n* = 4) and tumor-free mice (*n* = 7). **c** Reduced NET release by ᴳneutrophils (isolated from *n* = 5 mice) in comparison to lung neutrophils from MOPC-bearing (*n* = 10) and tumor-free mice (*n* = 6). **d** Impaired actin polymerization of ᴳneutrophils (isolated from *n* = 4 mice) in comparison to lung neutrophils from MOPC-bearing (*n* = 5) and tumor-free mice (*n* = 5) after lipopolysaccharide stimulation, determined by phalloidin–FITC staining. **e** Representative image of reduced NET release by ᴳneutrophils (orange, histone 1; blue, DAPI; scale bar = 65 μm). **f** Elevated Mpo protein expression in ᴳneutrophils (isolated from *n* = 4 mice) in comparison to lung neutrophils from MOPC-bearing (*n* = 5). **g** Elevated Mpo gene expression in ᴳneutrophils (isolated from *n* = 4 mice) in comparison to lung neutrophils from MOPC-bearing (*n* = 3) and tumor-free mice (*n* = 4). **h** Increased mitochondrial abundance in ᴳneutrophils (isolated from *n* = 5 mice) in comparison to lung neutrophils from MOPC-bearing (*n* = 5) and tumor-free mice (n = 4).

**i** Representative image of increased mitochondrial abundance in ᴳneutrophils (red, PKH26 membrane dye; green, MitoSpy; blue, Hoechst; scale bar = 10 μm). **j** Elevated ROS production in unstimulated ᴳneutrophils (isolated from *n* = 7 mice) in comparison to lung neutrophils from MOPC-bearing (*n* = 4) and tumor-free mice (*n* = 6), measured with 123-DHR. **k** Elevated ROS production in P. aeruginosa–stimulated ᴳneutrophils (isolated from *n* = 4 mice) in comparison to lung neutrophils from MOPC-bearing (*n* = 4) and tumor-free mice (*n* = 4), measured with 123-DHR. **l** Representative histograms of ROS production (light green, unstimulated neutrophils; dark green, P. aeruginosa–stimulated neutrophils; light brown, unstimulated ᴳneutrophils; dark brown, P. aeruginosa–stimulated ᴳneutrophils). **m**, **n** Mmp9 levels. **m** Increased Mmp9 in lung homogenates in ᴳMOPC-bearing mice (*n* = 4) in comparison to lungs from MOPC-bearing (*n* = 4) and tumor-free mice (*n* = 4). **n** Increased Mmp9 in supernatants after 24 h culture of ᴳneutrophils (isolated from *n* = 4 mice) in comparison to lung neutrophils from MOPC-bearing (*n* = 4) and tumor-free mice (*n* = 3). **o** Elevated Mmp9 gene expression in ᴳneutrophils (isolated from *n* = 4 mice) in comparison to lung neutrophils from MOPC-bearing (*n* = 4) and tumor-free mice (*n* = 4). Co control tumor-free mice, MOPC mice bearing non-G-CSF-producing tumors, ᴳMOPC ice bearing G-CSF-producing tumors, NETs neutrophil extracellular traps, Mpo myeloperoxidase, ROS reactive oxygen species, Mmp9 matrix metalloproteinase 9, CFU colony-forming units. Data are presented as means with individual values (biological replicates) shown. Statistical tests: Kruskal–Wallis for multiple-group comparisons with Bonferroni correction, and two-sided Mann–Whitney for two-group comparisons. *$p$ < 0.05, **$p$ < 0.01, ***$p$ < 0.001; #$p$ < 0.1. Source data and exact *p*-values are provided as a Source Data file.

---

impaired NET formation (Fig. 4i–k), and inability to produce ROS in response to bacteria (Fig. S8d).

Importantly, a similar trend can be observed in cancer patients: high production and release of G-CSF by the tumor microenvironment is associated with the accumulation of CD62Lˡᵒʷ aged neutrophils in tissues (Fig. 4l, m). Such CD62Lˡᵒʷ neutrophils show high cytotoxic properties, with spontaneous ROS production (Fig. 4n). Moreover, these cells show an exhausted phenotype with diminished antibacterial activity in response to *P. aeruginosa*, reduced NET production (Fig. 4o), and decreased phagocytosis (Fig. 4p).

Notably, we did not reveal signs of dysregulation of immunoactivation or immunosuppression-associated proteins in peripheral (lung) neutrophils associated with ᴳMOPC tumors (Supplementary Data 1), as well as in tumor neutrophils (Fig. S13) in the murine model.

### Neutrophils differentiated in the presence of G-CSF phenocopy ᴳneutrophils

Next, we were interested in whether the long-term exposure to tumor-derived G-CSF impacts neutrophil development and maturation, in addition to the observed modulation of mature neutrophil phenotype. Therefore, we developed the system of in vitro maturation of bone marrow-derived progenitors for mechanistic studies (modified from our previous studies[16]) (scheme of the experiment Fig. 5a).

Progenitors matured in the long-term presence of tumor-conditioned medium containing high levels of G-CSF (ᴳMOPC, Fig. 2a) showed accelerated differentiation into mature polymorphonuclear Ly6G⁺ neutrophils (Fig. 5b, c, Fig. S14). Such neutrophils phenocopied tumor-induced ᴳneutrophils, showing decreased granularity (Fig. 5d), elevated aged CD62Lˡᵒʷ population (Fig. 5e), decreased CD11b expression (Fig. 5f), higher spontaneous ROS production (Fig. 5g), and decreased phagocytic capacity (Fig. 5h).

To prove the essential role of tumor-secreted G-CSF in the suppression of neutrophil bactericidal activity, we block the G-CSF receptor during the entire neutrophil maturation using monoclonal antibodies. In line with our hypothesis, we observed impaired accumulation of aged CD62Lˡᵒʷ cells and suppressed ROS production. Similarly, after blocking downstream G-CSF signaling using STAT3 inhibitor LLL12, we observed partial restoration of neutrophil properties, namely inhibited neutrophil degranulation and lower accumulation of CD62Lˡᵒʷ cells, confirming the key role of G-CSF/ G-CSFR axis

in the tumor-induced dysregulation of neutrophil maturation and functions (Fig. S15).

To exclude that observed changes are simply the consequence of delayed apoptosis in ᴳneutrophils, we inhibited Caspase 3 using QVD-OPh and assessed neutrophil activity. Importantly, the treatment did not induce any changes observed after prolonged G-CSF exposure (Fig. S16).

### Long-term exposure to G-CSF persistently reprograms neutrophil progenitors

Development of neutrophils from hematopoietic stem cells in the bone marrow takes approximately 14 days[17]; therefore, we hypothesized that the clinical consequence of cancer would be tumor-associated modulation of granulopoiesis and impaired neutrophil functionality, even after surgical removal of the tumor. To test this, we performed an in vitro granulopoiesis assay, using bone marrow progenitors and compared neutrophil development in the presence and absence of tumor-derived factors (scheme of the experiment Fig. 5a). Indeed, such neutrophils exposed to ᴳMOPC-conditioned medium show decreased apoptosis (Fig. 5i), elevated degranulation (Fig. 5j), lower CD11b expression (Fig. 5k), elevated spontaneous production of ROS (Fig. 5l), and downregulated phagocytic capacity (Fig. 5m), as compared to neutrophils that mature in control conditions.

Next, to assess the effect of G-CSF on human neutrophil progenitors, we analyzed Gene Expression Omnibus databases GSE11247,[18] Fig. S17, in which gene expression profiles of CD133⁺ circulating stem cells mobilized by CXCR4 inhibitor AMD3100 alone, or in combination with G-CSF, were investigated, and compared to our proteomics data. Among the pathways significantly upregulated by G-CSF were those responsible for defense response and leukocyte activation (Fig. S17a), including beta-actin (Actb), commonly regulated (Fig. S17b). Downregulated pathways include those involved in cell adhesion and migration, regulation of cell death, and response to oxygen-containing compounds (Fig. S17a), with CD11b (Itgam) and cytochrome b (Cybb) commonly downregulated (Fig. S17c). This indicates that the changes induced by G-CSF in stem cells are persistent during their development and also present at the mature neutrophil level.

These data support the hypothesis that the changes observed in ᴳneutrophils are induced already at the neutrophil progenitor level, and could be persistent in neutrophils even after the removal of G-CSF,

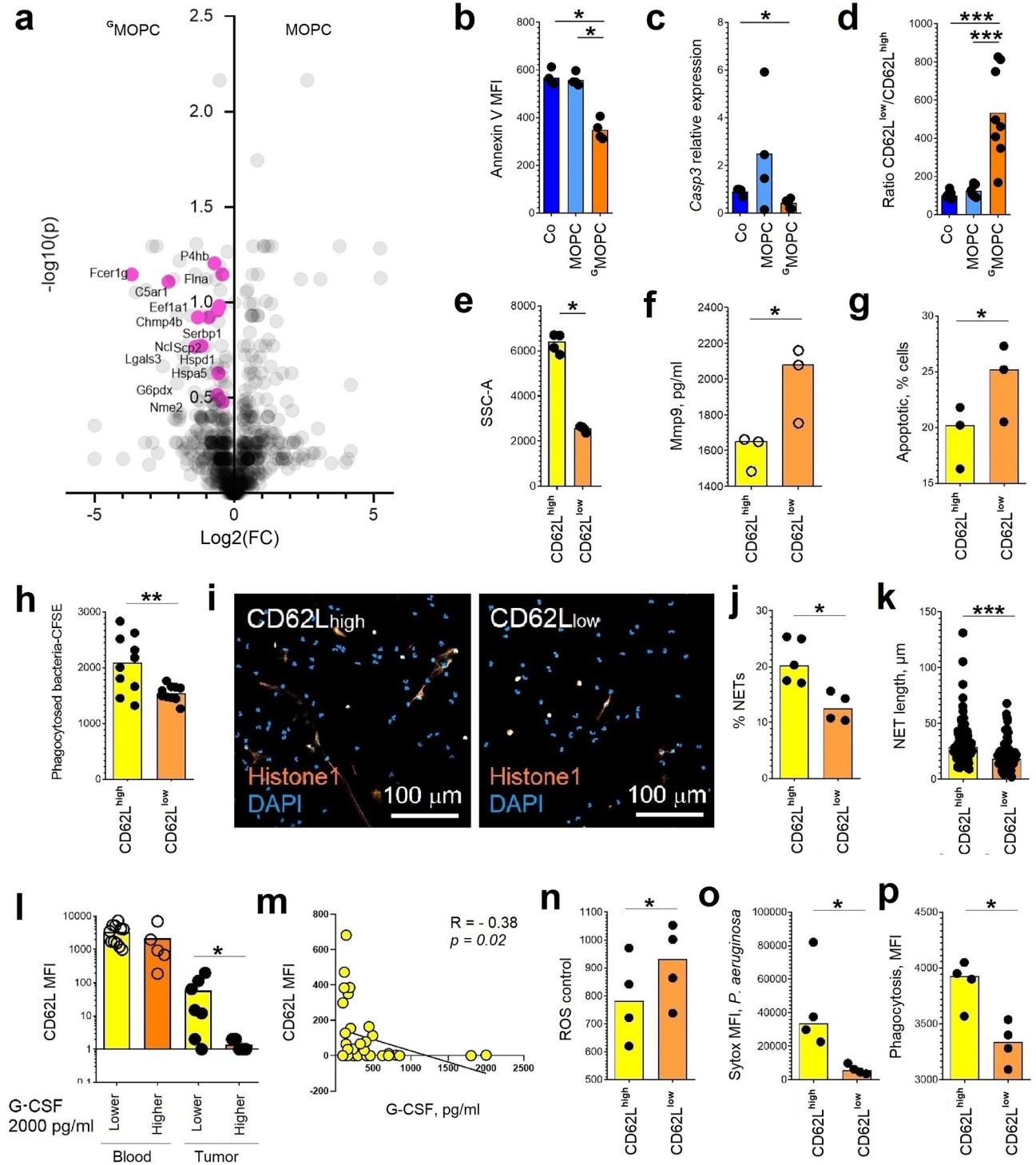

and be responsible for a prolonged susceptibility to bacterial infections.

## Therapeutic targeting of G-CSF signaling rescues neutrophil antibacterial functionality

G-CSF signaling pathway seems to be involved in the inhibition of neutrophil antibacterial properties; therefore, we hypothesized that targeting this pathway would have beneficial effects on their functionality. Previously, we observed that Nampt/NAD+ is essential for the activity of G-CSF signaling.[19] Although we could not measure any alteration of Nampt expression in [G]neutrophils on gene and protein level (Fig. S18a, b), we confirmed accumulation of NAD+ in [G]MOPC-

bearing mice (Fig. 6a), suggesting elevated activity of Nampt. As NAD+ accumulation can be due to decreased NAD+ consumption, we analyzed the abundance of major NAD-consuming enzymes in the whole lung proteome. The PARP/CD38/sirtuins pathway was not significantly altered in the lung tissue. The efficacy of inhibition of NAMPT activity in reversing neutrophil defects provides functional evidence for NAMPT's involvement (Fig. S18c).

To mechanistically assess the role of Nampt/NAD+ axis in neutrophil hyperactivation and tissue toxicity, we used Nampt inhibitor (FK866) as described before.[19] Indeed, inhibition of G-CSF downstream signaling reduced spontaneous ROS production by neutrophils maturing in the presence of tumor-derived factors (Fig. 6b).

**Fig. 4 | Chronic exposure to tumor-derived G-CSF alters $^G$neutrophil ageing and promotes accumulation of tissue-toxic CD62L$^{low}$ subpopulations. a** Mice were injected subcutaneously with MOPC or $^G$MOPC cells ($n = 5$ mice in each group). On day 14, animals were sacrificed, lungs were collected, viable Ly6G$^+$ neutrophils (gating strategy Fig. S3a) were isolated and analyzed by proteomics. Volcano plot of proteins differentially expressed in $^G$neutrophils according to SNR > 1, highlighting upregulated proteins involved in the regulation of neutrophil ageing and apoptosis. **b**–**k** Isolated viable Ly6G$^+$ lung $^G$neutrophils stratified by CD62L surface expression were evaluated in vitro in the absence or presence of Pseudomonas aeruginosa (MOI 10). **b** Decreased apoptosis in $^G$neutrophils (isolated from $n = 4$ mice) in comparison to lung neutrophils from MOPC-bearing ($n = 4$) and tumor-free mice ($n = 4$). **c** Reduced Casp3 gene expression in $^G$neutrophils (isolated from $n = 7$ mice) in comparison to lung neutrophils from MOPC-bearing ($n = 4$) and tumor-free mice ($n = 4$). **d** Accumulation of CD62L$^{low}$ $^G$neutrophils in lungs in comparison to lung neutrophils from MOPC-bearing and tumor-free mice ($n = 8$ mice in each group). **e** Increased degranulation of CD62L$^{low}$ $^G$neutrophils in comparison to CD62L$^{high}$ $^G$neutrophils (isolated from $n = 4$ mice) (SSC signal). **f** Elevated Mmp9 release by CD62L$^{low}$ $^G$neutrophils in comparison to CD62L$^{high}$ $^G$neutrophils (isolated from $n = 3$ mice). **g** Increased cytotoxicity of CD62L$^{low}$ $^G$neutrophils in comparison to CD62L$^{high}$ $^G$neutrophils (isolated from $n = 3$ mice) toward epithelial tumor cells. **h** Reduced phagocytic capacity of CD62L$^{low}$ $^G$neutrophils in comparison to CD62L$^{high}$ $^G$neutrophils (isolated from $n = 10$ mice). **i** Representative image showing reduced NET release by CD62L$^{low}$ $^G$neutrophils (orange, histone 1; blue, DAPI; scale bar 100 μm). **j** Decreased proportion of NET-producing cells among CD62L$^{low}$ $^G$neutrophils ($n = 4$) in comparison to CD62L$^{high}$ neutrophils (isolated from $n = 5$ mice). **k** Reduced NET length released by CD62L$^{low}$ $^G$neutrophils ($n = 61$ NETs) in comparison to CD62L$^{high}$ $^G$neutrophils ($n = 104$ NETs). **l** G-CSF release by HNSCC (in tumor-conditioned medium) negatively correlates with CD62L expression on tumor-associated neutrophils ($n = 12$ tumors). **m** In HNSCC patients, oral G-CSF concentrations negatively correlate with CD62L expression on salivary CD66b$^+$ viable neutrophils ($n = 29$ oral rinses). **n**–**p** Circulating neutrophils were isolated from healthy donors, CD62L$^{high}$ and CD62L$^{low}$ subsets were compared. **n** Elevated spontaneous ROS production in CD62L$^{low}$ neutrophils in comparison to CD62L$^{high}$ neutrophils isolated from $n = 4$ healthy volunteers. **o** Reduced NET generation in CD62L$^{low}$ neutrophils in comparison to CD62L$^{high}$ neutrophils isolated from $n = 4$ healthy volunteers. **p** Reduced phagocytic capacity of CD62L$^{low}$ neutrophils in comparison to CD62L$^{high}$ neutrophils isolated from $n = 4$ healthy volunteers. G-CSF granulocyte colony-stimulating factor, Co control tumor-free mice, MOPC mice bearing non-G-CSF-producing tumors, $^G$MOPC mice bearing G-CSF-producing tumors, Casp3 caspase-3, SSC side scatter, Mmp9 matrix metalloproteinase, NETs neutrophil extracellular traps, ROS reactive oxygen species, CFU colony-forming units. Data are shown as means with individual values (biological replicates). Statistical tests: Kruskal–Wallis with Bonferroni correction for multiple-group comparisons and two-sided Mann–Whitney for two-group comparisons (**b**–**l**); Spearman correlation (**m**); paired Student's t-test (**n**–**p**). *$p < 0.05$, **$p < 0.01$, ***$p < 0.001$. Source data and exact $p$-values are provided as a Source Data file.

---

To confirm the involvement of Nampt in the G-CSF-driven impairment of neutrophil functionality, we performed in vitro maturation assays using the alternative Nampt inhibitor OT-82 during neutrophil differentiation from progenitor cells. OT-82 treatment normalized neutrophil differentiation rate, phenotype, and function, including restoration of CD62L expression, reduction of spontaneous ROS production, and improved phagocytic capacity, further supporting the role of Nampt activity in neutrophil reprogramming (Fig. S19a).

To determine whether the effects of NAMPT inhibition on neutrophil maturation and function can be bypassed by direct supplementation of the NAD$^+$ pathway, we conducted in vitro maturation assays using bone marrow progenitors in the presence of the NAMPT inhibitor FK866, with and without the addition of nicotinamide mononucleotide (NMN). Our results demonstrated that NMN supplementation effectively bypassed the effects of NAMPT inhibition, specifically by accumulation of aged CD62L$^{low}$ CD11low phenotype, with elevated production of ROS (Fig. S19b). These findings confirm that the observed effects of FK866 are specifically due to disruption of the NAD$^+$ biosynthetic pathway and can be reversed by providing NMN as a downstream metabolite.

Next, we assessed the effect of blocked G-CSF on the susceptibility to infection in vivo. In agreement with our hypothesis, treatment of $^G$MOPC-bearing mice with FK866 decreased lung infiltration with CD62L$^{low}$ neutrophils (Fig. 6c, d). Moreover, such neutrophils show a distinct pro-apoptotic phenotype (Fig. 6e–g), lower tissue-toxic potential with decreased spontaneous ROS production (Fig. 6h) and MMP9 release (Fig. 6i), but no changes in phagocytosis (Fig. 6j). In line with this, mice treated with FK866 demonstrated improved bacterial clearance (Fig. 6k) and clinical performance (Fig. 6l), in comparison to untreated mice.

Moreover, treatment of tumor-bearing mice with oral Nampt inhibitor OT-82 (20 mg/kg orally, once daily for three consecutive days followed by four days of rest, repeated for two weeks) resulted in a similar repolarization of neutrophils, characterized by reduced accumulation of dysfunctional, tissue-toxic neutrophil subsets in lungs (Fig. S19c). These effects were consistent with the phenotypic and functional changes previously reported with FK866.

This implies the importance of therapeutic approaches aiming at the normalization of the G-CSF axis in cancer individuals to prevent neutrophil tissue toxicity, but at the same time to support antibacterial properties of such neutrophils to minimize the susceptibility of patients to bacterial infections.

To sum up (Fig. 7), chronic exposure to tumor-derived G-CSF not only enhances granulopoiesis and the release of harmful cytotoxic neutrophils from the bone marrow but also results in the local retention of senescent exhausted neutrophils with diminished antibacterial properties in the lung. This leads to significant lung tissue damage and enhanced lung colonization by bacteria.

## Discussion

Cancer patients often suffer from recurrent bacterial infections that have a fatal impact on their morbidity and mortality. One of the reasons responsible for this phenomenon could be G-CSF, which is chronically released by the growing tumor. Long-term exposure to G-CSF, in contrast to short-term treatment, induces a dysfunctional, exhausted state of neutrophils, with elevated cytotoxic activity and diminished antibacterial responses. This leads to tissue damage and impaired bacterial clearance, and thus to prolonged infections.

Neutrophils are the key players orchestrating antibacterial immunity,[5] and are reported to be significantly affected by cancer-released factors.[10] One of such factors is granulocyte colony-stimulating factor (G-CSF),[20] which exerts ambivalent effects on neutrophil antibacterial activity. Importantly, while short-term G-CSF treatment stimulates antibacterial activity of neutrophils,[14] the evidence from the clinical trials reports the lack of protective effect[21,22] or even immuno-inhibitory properties[23,24] of long-term G-CSF treatment.

Short-term treatment with G-CSF was previously shown to induce the mobilization of neutrophils and to stimulate the antibacterial potential of circulating neutrophils.[14] Therefore, the role of G-CSF treatment for chemotherapy-induced febrile neutropenia and prevention of sepsis is hard to underestimate.[25]

Under physiological conditions, G-CSF is primarily cleared by neutrophils and neutrophil precursors, meaning that clearance from the circulation is a self-regulating process. After binding of G-CSF to its receptor (G-CSFR), G-CSF/G-CSFR complex is internalized and degraded.[26] In the context of cancer, tumor tissue constantly releases G-CSF; therefore, self-regulation of G-CSF concentration is not possible, resulting in aberrant neutrophil functionality. Our data indicate that the high concentration of G-CSF found in the oral rinse of HNSCC patients (varying between 100-1000 pg/ml in saliva) is most likely

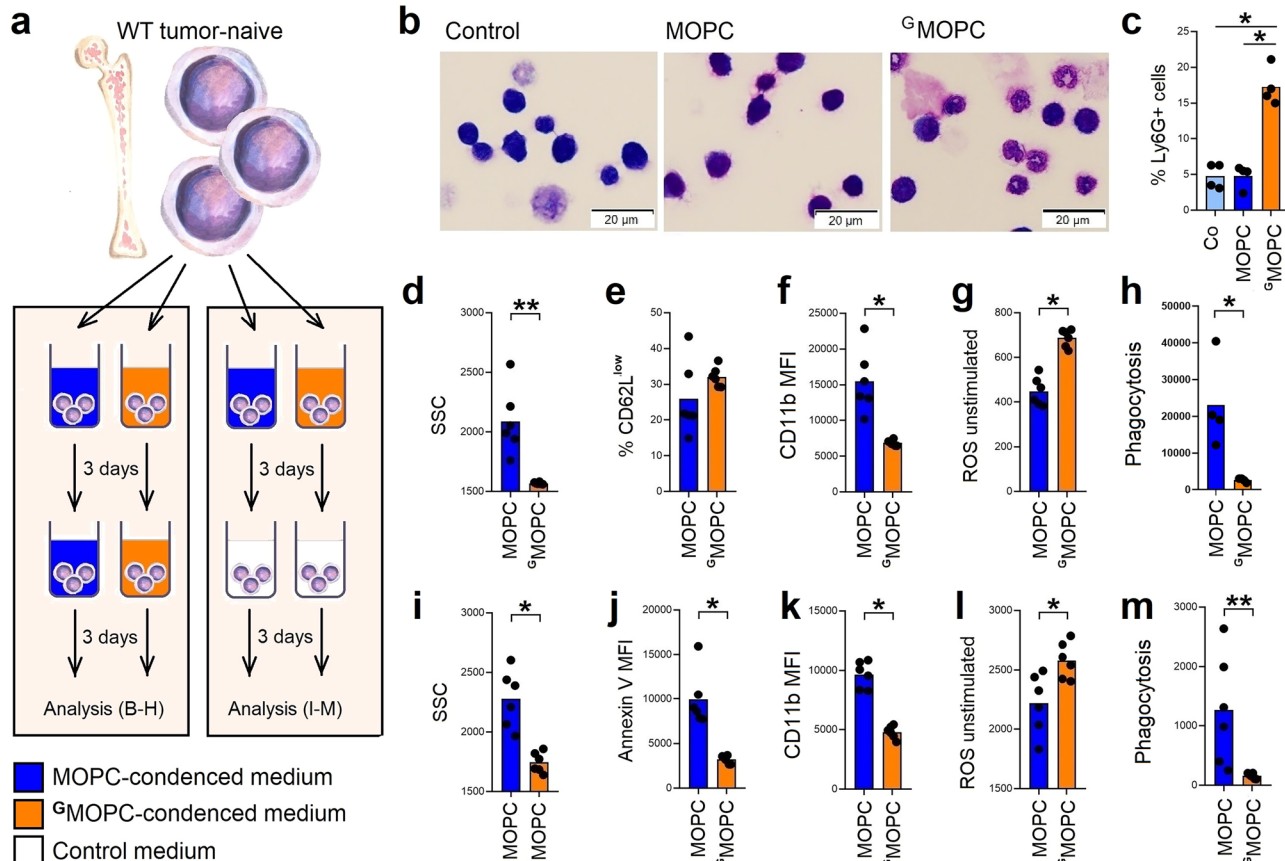

**Fig. 5 | Tumor-derived G-CSF induces persistent functional changes in neutrophil progenitors in vitro. a** Experimental design. Bone marrow progenitors (CD3e⁻CD45R⁻NK1.1⁻CD11b⁻Ly6G⁻Ter119⁻) were isolated by immunomagnetic negative selection (modified from ref. 16). Cells were cultured with MOPC- or ᴳMOPC-conditioned medium for 6 days (a, left), or for 3 days followed by 3 days in control medium without stimulation (a, right). On day 6, the phenotype and function of newly differentiated Ly6G⁺ neutrophils (gating strategy Fig. S3a) were assessed. **b–h** Bone marrow progenitors cultured with MOPC- or ᴳMOPC-conditioned medium for 6 days were analyzed for morphology, phenotype, and function. **b** Representative Giemsa-stained cytospins (scale bar = 20 μm). **c** ᴳMOPC-conditioned medium induced elevated differentiation of progenitors from $n = 4$ mice into Ly6G⁺ neutrophils in comparison to MOPC or –conditioned medium or control medium. **d** ᴳMOPC-conditioned medium enhanced degranulation of Ly6G+ matured from progenitors isolated from $n = 6$ mice, in comparison to MOPC or –conditioned medium. **e** ᴳMOPC-conditioned medium increased the proportion of CD62Lˡᵒʷ Ly6G⁺ neutrophils matured from progenitors isolated from $n = 6$ mice, in comparison to MOPC or –conditioned medium. **f** ᴳMOPC-conditioned medium reduced CD11b expression on Ly6G⁺ neutrophils matured from progenitors isolated from $n = 6$ mice, in comparison to MOPC or –conditioned medium. **g** ᴳMOPC-conditioned medium elevated unstimulated ROS production of Ly6G⁺ neutrophils matured from progenitors isolated from $n = 6$ mice, in comparison to MOPC–conditioned medium. **h** ᴳMOPC-conditioned medium decreased phagocytosis of Ly6G⁺ neutrophils matured from progenitors isolated from $n = 4$ mice, in

comparison to MOPC–conditioned medium. **i–m** Bone marrow progenitors were exposed to MOPC- or ᴳMOPC-conditioned medium for 3 days, followed by 3 days in control medium; newly differentiated Ly6G⁺ neutrophils were analyzed. **i** Initial ᴳMOPC-conditioned medium exposure increased degranulation of Ly6G+ neutrophils matured from progenitors isolated from $n = 6$ mice, in comparison to MOPC–conditioned medium. **j** Initial ᴳMOPC-conditioned medium exposure increased apoptosis of Ly6G⁺ neutrophils matured from progenitors isolated from $n = 6$ mice, in comparison to MOPC–conditioned medium. **k** Initial ᴳMOPC-conditioned medium exposure reduced CD11b expression on Ly6G+ neutrophils matured from progenitors isolated from $n = 6$ mice, in comparison to MOPC–conditioned medium. **l** Initial ᴳMOPC-conditioned medium exposure elevated unstimulated ROS production by Ly6G+ neutrophils matured from progenitors isolated from $n = 6$ mice, in comparison to MOPC–conditioned medium **m** Initial ᴳMOPC-conditioned medium exposure reduced phagocytosis of Ly6G+ neutrophils matured from progenitors isolated from $n = 6$ mice, in comparison to MOPC–conditioned medium. Co control (baseline) progenitor culture, MOPC cells incubated with MOPC-conditioned medium, ᴳMOPC cells incubated with ᴳMOPC-conditioned medium, SSC side scatter, ROS reactive oxygen species. Data are presented as means with individual values shown. Statistical tests: Kruskal–Wallis for multiple-group comparisons with Bonferroni correction; two-sided Mann–Whitney for two-group comparisons. *$p < 0.05$, **$p < 0.01$, ***$p < 0.001$; #$p < 0.1$. Source data and exact $p$-values are provided as a Source Data file.

derived directly from the tumor microenvironment rather than from systemic circulation (where the levels are around 10 pg/ml).

Neutrophils from HNSCC patients in comparison to neutrophils from healthy individuals [GSE79404[27]] display molecular signatures of active G-CSF signaling (JAK-STAT signaling pathway hsa04630), indicating functional exposure to G-CSF. Previous studies, including our own work,[19] have documented heterogeneity in plasma G-CSF levels among HNSCC patients, particularly when stratified by disease stage. This suggests that systemic G-CSF may be elevated in subsets of patients or at specific disease stages, which could contribute to neutrophil reprogramming in those contexts. The tumor

microenvironment can create high local concentrations of G-CSF, particularly within the tumor and draining tissues. Neutrophil progenitors trafficking through or residing in these areas[28] can be directly influenced by these elevated cytokine levels, even if systemic levels remain low. Another aspect of this is that G-CSF may reach the bone marrow or peripheral tissues not only as a soluble factor in plasma but also encapsulated within extracellular vesicles (EVs). There is direct evidence that extracellular vesicles (EVs) can accumulate in bone marrow following their release from distant tissues.[29] This accumulation in BM during inflammation is described as notable and significant, beyond mere passive distribution in plasma, with enhanced functional

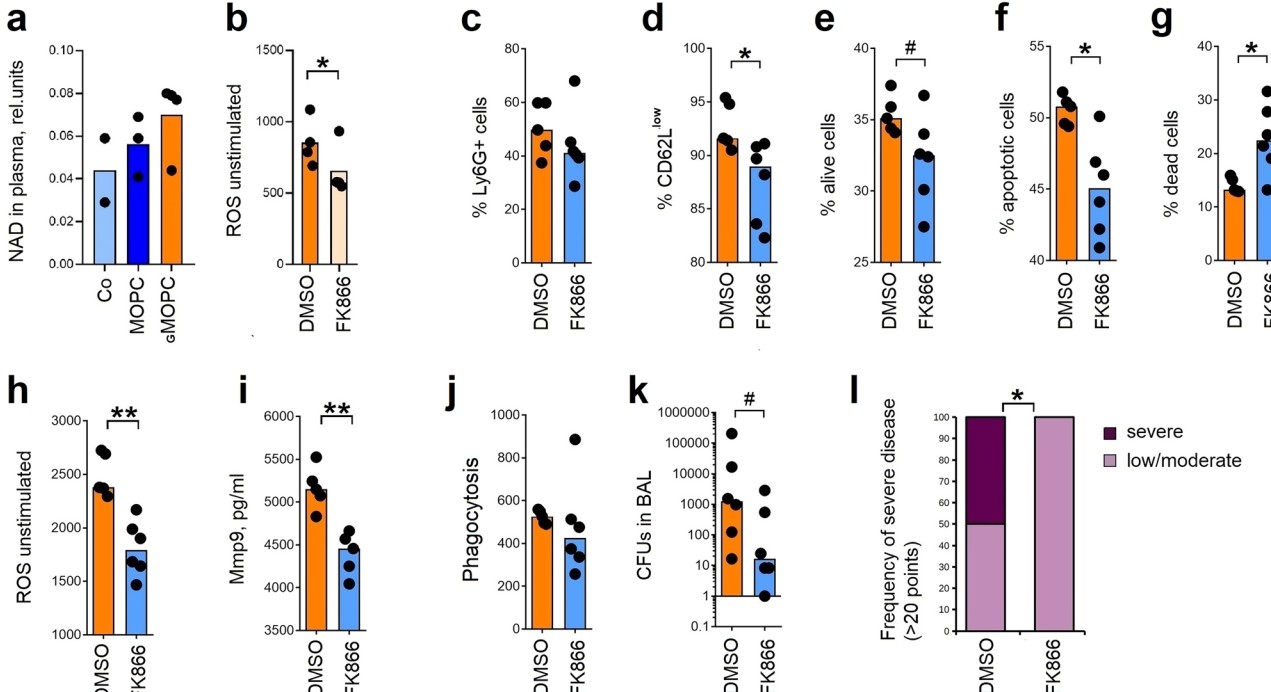

**Fig. 6 | Inhibition of G-CSF receptor downstream signaling abrogates tissue toxicity of G-neutrophils and improves bacterial clearance. a** Mice were injected subcutaneously with PBS (control, Co), MOPC, or GMOPC cells. On day 14, animals were sacrificed, and blood was collected. Plasma nicotinamide adenine dinucleotide (NAD) levels were elevated in GMOPC-bearing mice (n = 4) in comparison to MOPC-bearing (n = 3) and tumor-free mice (n = 2). **b** Bone marrow progenitors were cultured for 6 days in GMlOPC-conditioned medium with or without the NAMPT inhibitor FK866. Inhibition of NAMPT decreased unstimulated reactive oxygen species (ROS) production in newly differentiated Gneutrophils (n = 4) in comparison to untreated Gneutrophils (n = 4) in biological replicates (gating strategy Fig. S3a). **c–l** GMOPC-bearing mice were treated intraperitoneally with FK866 every other day. On day 14, animals were sacrificed, and neutrophil (gating strategy Fig. S3a) phenotype and function were analyzed in lung single-cell suspensions. **c** FK866 reduced lung infiltration by Gneutrophils (n = 6) in comparison to untreated mice (n = 5). **d** FK866 prevented accumulation of CD62Llow Gneutrophils (n = 6) in comparison to untreated mice (n = 5). **e–g** FK866 increased Gneutrophil cell death (n = 6) in comparison to untreated mice (n = 5). **h** FK866 decreased unstimulated ROS production in Gneutrophils (n = 6) in comparison to untreated

mice (n = 5). **i** FK866 normalized Pseudomonas aeruginosa–stimulated Mmp9 release by Gneutrophils (n = 5) in comparison to untreated mice (n = 5). **j** FK866 had no effect on the phagocytic capacity of Gneutrophils (n = 6) in comparison to untreated mice (n = 5). **k, l** GMOPC-bearing mice were treated with FK866 every other day. On day 14, mice were infected intratracheally with P. aeruginosa and sacrificed 18 h later. **k** FK866 enhanced bacterial clearance in the lower respiratory tract (n = 6) in comparison to untreated mice (n = 6). **l** FK866 improved clinical performance of infected mice (n = 6) in comparison to untreated mice (n = 6). Co control tumor-free mice, MOPC mice bearing non-G-CSF-producing tumors, GMOPC mice bearing G-CSF-producing tumors, NAD nicotinamide adenine dinucleotide, ROS reactive oxygen species, Mmp9 matrix metalloproteinase-9, CFU colony-forming units, NAMPT nicotinamide phosphoribosyltransferase, BAL bronchoalveolar lavage. Data are presented as means with individual values (biological replicates) shown. Statistical tests: Kruskal–Wallis for multiple-group comparisons with Bonferroni correction, and two-sided Mann–Whitney for two-group comparisons (**a–k**); Chi-square (**l**). *p < 0.05, **p < 0.01, ***p < 0.001; #p < 0.1. Source data and exact p-values are provided as a Source Data file.

consequences for hematopoietic and immune cell dynamics. Such vesicle-mediated transport can facilitate targeted cytokine delivery and may evade detection by conventional ELISA assays.

The impact of G-CSF on neutrophil activity apparently depends on the dose, duration, and underlying disease. In the context of hematopoietic stem cell mobilization by G-CSF and their further transplantation, impaired chemotaxis was observed, both in donor and in recipient neutrophils.[23,30,31] At the same time, reported changes of neutrophil functions are controversial: increased functionality in healthy donors,[31] decreased ROS and phagocytosis in transplant recipients,[32] or no changes in ROS and phagocytosis in both.[23] Our experiments show diminished neutrophil functionality after long-lasting exposure to tumor-derived G-CSF in non-neutropenic conditions, associated with impaired antibacterial responses (NET formation and phagocytosis, due to impaired actin cytoskeleton reorganization). Moreover, our data demonstrate a clear morphological cause (lack of proteins regulating actin polymerization) responsible for the impairment of Gneutrophil motility and associated functions, including phagocytosis and NET formation, in addition to functional exhaustion.[23]

Neutrophils represent the first line defenders in acute inflammatory responses.[33] Aged/exhausted CD62Llow neutrophils have been

shown to contribute to sterile vascular injury and thrombosis in the model of fungal infection, despite impaired actin cytoskeleton.[34] We demonstrated that tumor-derived G-CSF not only prolonged survival, but also reprogrammed neutrophils during granulopoiesis, which resulted in their aged, exhausted phenotype, with impaired antimicrobial activity and elevated tissue toxicity. A typical marker of such aged neutrophils is downregulated CD62L (Selllow) surface expression. CD62L is highly expressed in young neutrophils, but is decreased on aged or activated neutrophils due to reduced gene expression or shedding.[35] G-CSF was shown to support shedding of CD62L,[36] therefore prolonged exposure to G-CSF in cancer can be responsible for the CD62Llow phenotype of neutrophils. Accumulation of such tissue-toxic neutrophils in organs, driven by tumor-derived G-CSF, can be responsible for acute respiratory distress syndrome, which is characteristic of G-CSF treatment.[37]

Several therapeutic strategies have already been tested to neutralize the adverse effects of G-CSF in neutrophils. Neutralizing anti-G-CSFR antibodies were shown to block G-CSF-induced neutrophilia, without inducing neutropenia, in non-human primates.[38] Moreover, anti-G-CSFR antibodies reduced neutrophilic inflammation during pneumococcal or influenza respiratory infections, without

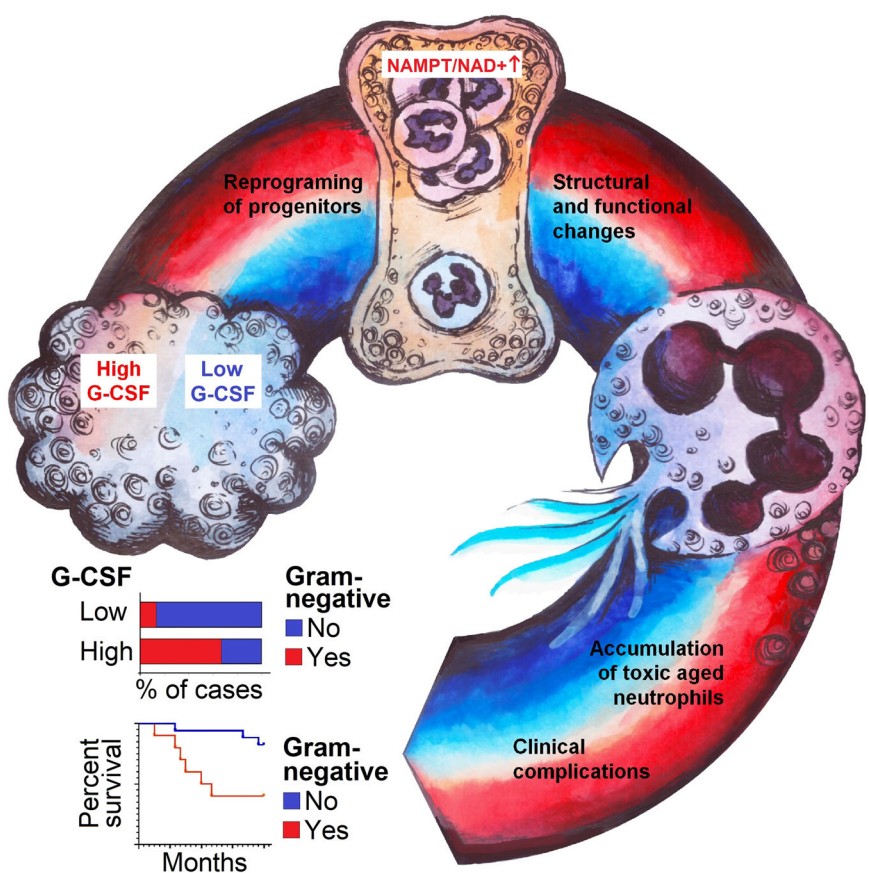

**Fig. 7 | Graphical abstract.** Chronic tumor-derived G-CSF–NAMPT signaling reprograms neutrophil progenitors, drives accumulation of tissue-toxic, infection-permissive neutrophils and is associated with persistence of Gram-negative pathogens and worse outcome in cancer. G-CSF granulocyte colony-stimulating factor, NAMPT nicotinamide phosphoribosyltransferase, NAD+ nicotinamide adenine dinucleotide.

compromising bacterial clearance[39] and increased neutrophil CD62L expression, reverting them to an active antibacterial phenotype.[40]

Neutrophil-released ROS is one of the most powerful cytotoxic agents, triggering tissue damage. In addition, it supports the formation of biofilms by various bacterial species, including *P. aeruginosa*, by inducing the overproduction of capsule-like exopolysaccharide alginate (so-called mucoid conversion).[41] High ROS production can be a result of upregulated downstream G-CSF signaling, which supports salvage $NAD^+$ synthesis through upregulation of nicotinamide phosphoribosyltransferase (NAMPT).[42] Elevated $NAD^+$ production coincides with the engagement of oxidative phosphorylation, as a result of increased oxygen availability.[43] Previously, NAMPT was demonstrated to mitigate colitis severity by supporting redox-sensitive activation of phagocytosis in inflammatory macrophages,[44] nevertheless, we demonstrated that in particular cases (high G-CSF production by tumor and elevated tissue toxicity of neutrophils), inhibition of NAMPT can have beneficial effects for the host.

We hypothesized here that reprogramming of neutrophils by tumor-derived G-CSF occurs already at the progenitor stage. In agreement, elimination of tumor-derived G-CSF from the system after prolonged neutrophil exposure to tumor-conditioned medium failed to fully restore their phenotype and properties (ROS, phagocytosis), demonstrating the crucial impact of the early education of neutrophil progenitors for their functionality. Observations from transplantology, where the persistence of aberrant G-CSF-stimulated neutrophils was observed in recipients in certain cases up to several weeks after stem cell transplantation,[23,32] confirm the long-lasting effect of neutrophil progenitor reprogramming. Hence, tumor-induced reprogramming of myeloid progenitors in hematopoietic organs can be responsible for

their prolonged suppressed bactericidal activity. It is especially important because clinically, the majority of bacterial complications occur within 30 days after surgery, with almost half of the cases occurring after hospital discharge.[45] Thus, developing ways to monitor neutrophil functionality in patients after oncologic surgery should be a high priority to identify patients at risk.

Early non-cancer mortality, usually defined as 90-day mortality after diagnosis or treatment initiation, is a main driver of overall mortality in the HNSCC population. Prevalence of early non-cancer death has been shown to be around 5% in several cohorts, with about 30% of cases being caused by pulmonary infection or bloodstream infection[46] affecting patients treated with primary surgery[47] or (chemo-)radiation[48] equally. Given that a significant proportion of these patients are expected to be cured from their tumor, non-cancer mortality is an unacceptable obstacle to achieving good patient outcomes.

Unfavorable changes in neutrophil functionality due to prolonged G-CSF exposure may worsen the prognosis of cancer patients. The prognostic role of G-CSF expression in tumor tissue is also already known.[19] Here, we observed a worse prognosis of HNSCC patients with elevated G-CSF levels, which was associated with the presence of Gram-negative pathogens and increased risk of bacterial complications. The aerodigestive tract, especially the oral cavity, serves as a reservoir of bacteria, which can then spread with saliva. The incidence of *P. aeruginosa* is twice as high in people with untreated head and neck cancer, in comparison to healthy individuals,[4] indicating that cancer-derived factors influence immune antibacterial responses, independently of the treatment. Further anticancer therapies, due to their cytostatic and thus immunosuppressive effects, as well as due to the disruption of mucosal barriers, may allow the spreading of

persisting pathogens. Gram-negative pathogens, including *Pseudomonas*, *Escherichia*, or *Klebsiella spp*, represent the major reason for local and systemic infectious complications after surgery for multiple tumor entities, including HNSCC.[45] Therefore, prevention of bacterial complications and treatment remains a great challenge in cancer patients. Additionally, infectious complications might lead to cancer progression if they interfere with indicated standard of care treatment, such as in the case of adjuvant radiation therapy. Delays in post-operative treatment have been repeatedly associated with worse oncologic outcome,[49] but the reasons for extended lags between surgery and the initiation of postoperative radiotherapy are unclear at this point. Even though these reasons will be plentiful and variable, it is rational to assume that infections, their treatment, and the time required to recuperate are responsible for a significant subset of delays. This highlights the need for early identification and treatment of patients at risk of infectious complications.

Here, we demonstrate that one of the reasons for bacterial persistence and spread in tumor-bearing hosts is severely impaired antibacterial properties of neutrophils, accompanied by their eminent tissue toxicity that is caused by tumor-derived G-CSF. Therefore, treatment modalities should be considered to neutralize the impact of cancer-related G-CSF stimulation of neutrophils in order to prevent tissue damage and bacterial persistence in damaged tissue. Prediction and early diagnostics of patient predisposition to bacterial complications would decrease possible therapy delay and thus improve patient survival.

## Methods
The research complies with all relevant ethical regulations

### Animal experiments
The animal experiments have been approved by the regulatory authorities LANUV (Das Landesamt für Natur, Umwelt und Verbraucherschutz Nordrhein-Westfalen), Germany. Our animal care and use protocols adhere to the regulations of das Deutsche Tierschutzgesetz (TierSchG) and follow FELASA recommendations. Mice were monitored daily throughout the experiment. Tumor progression and disease burden were assessed during routine daily checks by visual inspection and caliper measurement. For non-infected mice, the experimental endpoint was day 14 post-tumor implantation. For mice subjected to *Pseudomonas aeruginosa*-induced lower respiratory tract infection, infection was performed on day 14, and the experimental endpoint was day 15 post-tumor implantation. If, during the course of daily observation, any animal reached a humane endpoint (cumulative burden of 20 points (Supplementary Table 1), a tumor diameter of 1 cm, or developed an ulcerated tumor), the animal was immediately euthanized according to ethical guidelines. At no time did animals exceed these limits.

Research involving human material, or human data, was performed in accordance with the Declaration of Helsinki and was approved by the ethics committee of the University Hospital Essen, Germany (19-8599-BO, 16-7135-BO). Informed consent to participate in the study was obtained from participants.

When designing and implementing the research, we considered issues of equity, diversity, and inclusion. The study population was selected with attention to avoiding exclusion on the basis of [e.g., gender, ethnicity, socioeconomic status, age, ability], and recruitment strategies were intended to minimize barriers to participation. We report demographic information in aggregate form where appropriate to protect participant confidentiality, while recognizing the importance of transparency in representing the diversity of our sample.

### Cell lines
The murine oropharyngeal carcinoma cell line MOPC (C57BL/6-derived, HPV16 E6/E7⁻) was obtained from Dr. William Chad Spanos

and John H. Lee (Sanford Research/University of South Dakota, Sioux Falls, SD, US).[50] Employing CRISPR/Cas9-mediated targeted knock-in technology, we engineered murine HNSCC cell lines expressing low and high G-CSF levels. Elevated production of G-CSF by tumor cells ($^G$MOPC cell line) in cell culture conditioned medium was evaluated with ELISA according to the manufacturer's protocol. Cells were cultivated in a special medium (67% DMEM, 22% Hams F12 nutrient mix, 10% Fetal Bovine Serum, 1% penicillin-streptomycin, 0.5 μg/ml Hydrocortisone, 8.4 ng/ml Cholera Toxin, 5 μg/ml Transferrin, 5 μg/ml Insulin, 1.36 ng/ml Tri-Iodo-Thyronine, 5 μg/ml E.G.F.). During cultivation, cell lines were regularly tested for mycoplasma contamination with negative results. Cells were grown in a monolayer at 37 °C in a humidified incubator with 5% $CO_2$.

### Animals
C57BL/6JCrl mouse strain from our own breeding (University Hospital Essen), originally a JAX strain bred by Charles River Laboratory, was used for experiments. For the experiments, female littermates between 8 and 12 weeks were used. Mice were housed and bred under specific pathogen-free conditions in cages of up to 5 mice per cage, 12 h light/dark cycle at the animal facility of the University Hospital Essen. All animal experiments have been approved by the regulatory authorities LANUV (Das Landesamt für Natur, Umwelt und Verbraucherschutz Nordrhein-Westfalen, Germany). Our animal care and use protocols adhere to the regulations of German law according to das Deutsche Tierschutzgesetz (TierSchG) and follow the recommendations of the Federation of European Laboratory Animal Science Associations (FELASA).

The MOPC and $^G$MOPC cells were injected subcutaneously (s.c. 1 ×106 in 100 μl PBS) into the flank of C57BL/6 mice, as described previously.[16] Tumor-free animals from the same strain were used as control animals.

### NAMPT inhibition in vivo
Mice treatment with the NAMPT inhibitor FK866 25 mg/kg (dissolved in DMSO, with DMSO alone as a control) was performed by *i.p.* injection at day 0 and further at every second day. Alternatively, treatment with OT-82 (MedChemExpress) 20 mg/kg (dissolved in DMSO and corn oil, according to the manufacturer's protocol, with DMSO and corn oil alone as a control) orally, once daily for three consecutive days, followed by four days of rest, repeated for two weeks, was performed.

### Bacteria
*P. aeruginosa* strains that were used in this study: PA14 parental strain. Bacteria have been cultured in Luria-Bertani (LB) broth for 3 h to reach the early exponential phase, washed twice in PBS, the optical density of 100 μl suspension was measured in 96-well flat-bottom cell culture plates (Cellstar, Greiner Bio One International GmbH, Frickenhausen, Germany) at 600 nm using a microplate reader Synergy 2 (BioTek Instruments, Inc., Vermont, U.S.). OD 0.4 corresponds to a bacterial density of $5 \times 10^9$/ml, as determined by serial dilutions and colony-forming unit (CFU) assays. Bacteria concentration was adjusted to the desired values and verified by plating on 2% LB agar plates.

### Lower respiratory tract infection in mice
For intratracheal inoculation of *P. aeruginosa*, mice were anesthetized with Ketamin (belapharm GmbH & Co, Vechta, Germany) 100 mg/kg and Xylazin (Ceva Tiergesundheit GmbH, Düsseldorf, Germany) 10 mg/kg in 0.9% NaCl solution, intubated, and $2 \times 10^6$ CFUs of *P. aeruginosa* in sterile PBS (50 μl) were administered using the Minivent Mouse Ventilator type 845 (Harvard Apparatus, Massachusetts, U.S.) with stroke volume 150 μl and frequency 150 breaths/min. The control of the distribution of liquid in both lungs during intratracheal administration was performed prior to the experiments using Trypan blue (Sigma-Aldrich/Merck, Darmstadt, Germany). The adapted intratracheal method demonstrated accurate delivery and retention of *P. aeruginosa* in

lungs. Animals were monitored post-operatively in a heated box until ambulant and clinically normal. Mice were transferred to a clean box with food and water ad libitum and monitored for 20 h. To evaluate the clinical status of the mice, a severity scoring was performed according to the experiment-specific score sheet approved in our animal permission, based on the guidelines of the Deutsche Tierschutzgesetz (Supplementary Table 1). After 20 h, mice were sacrificed. Heparinized blood was collected via heart puncture, and plasma was prepared after centrifugation. Bronchoalveolar lavage (BAL) was collected after bronchial perfusion through the trachea with 1 ml of sterile PBS. BAL was plated in serial dilutions to estimate CFUs on 2% LB agar and examined after 24 h of incubation.

Blood was collected after sacrificing via heart puncture in heparinized tubes. Plasma was collected after centrifugation at $2000 \times g$, frozen at $-80\,°C$ until further analysis. Alternatively, white blood cells were collected from blood after threefold lysis of red blood cells with ACK buffer containing $NH_4Cl$ 150 mM, $KHCO_3$ 10 mM, $Na_2EDTA$ 0.1 mM.

NAD levels in plasma were measured using the NAD/NADH Assay Kit (Abcam), following the manufacturer's protocol.

**Histology.** For histological examination of lungs, mice were infected *i.t.* with *P. aeruginosa*. At a certain time point, mice were sacrificed, lungs perfused with Tissue-Tek O.C.T. Compound (Sakura Finetek, Japan) containing 5% paraformaldehyde, the lumen of the trachea was fixed with ligature; lungs were dissected and snap frozen at $-80\,°C$. 7 μm cryosections were fixed with ice-cold acetone, stained with hematoxylin-eosin, dried, and mounted with Neo-Mount (Merck, Darmstadt, Germany). The percentage of aerated area was calculated from histological images by subtracting the percentage of the section occupied by lung tissue (alveolar septa and inflammatory infiltrates) from the total section area, and expressing the remaining area as a percentage of the total, using ImageJ Fiji software.

**Collagen staining.** Frozen lung tissue samples embedded in O.C.T. compound were cryosectioned at a thickness of 7 μm. Sections were air-dried, fixed in ice-cold methanol for 2 min, and subjected to Van Gieson's staining to visualize collagen fibers. The working Van Gieson solution was prepared by mixing 5 ml of 1% aqueous acid fuchsin with 100 ml of saturated aqueous picric acid, resulting in a final acid fuchsin concentration of 0.05%. Sections were incubated in this staining solution for 2 min, then rinsed, dehydrated through graded ethanol, cleared in xylene, and mounted with Neo-Mount medium. Stained slides were examined and imaged by bright-field microscopy.

**Microscopy.** Microscopy was performed using Zeiss AxioObserver.Z1 Inverted Microscope with ApoTome Optical Sectioning equipped with filters for: DAPI, FITC, Alexa Fluor 488, GFP, DsRed, Cy3, or Olympus BX51 upright epifluorescence microscope. Images were processed with ZEN Blue 2012 software or CellSens Dimension software (Olympus), respectively, and analyzed with ImageJ.

**Assessment of neutrophil infiltration in lungs and tumors.** Lungs were collected as described above; organs from non-infected animals were used as a control. Lung tissue was digested using dispase 0.2 μg/ml, collagenase A 0.2 μg/ml, and DNase I 100 μg/ml (all Sigma-Aldrich/Merck, Darmstadt, Germany) solution in DMEM (Gibco, Life Technologies/Thermo Fisher Scientific, Massachusetts, U.S.) containing 10% FCS and 1% penicillin-streptomycin. Cells were meshed through 50 μm filters (Cell Trics, Partec, Sysmex Europe GmbH, Goerlitz, Germany) and erythrocytes lysed in ACK buffer containing $NH_4Cl$ 150 mM, $KHCO_3$ 10 mM, $Na_2EDTA$ 0.1 mM. Single-cell suspensions were stained with antibodies and reagents listed below.

**Isolation of lung neutrophils.** For the estimation of neutrophil functions, neutrophils were isolated from the lungs of non-infected mice. Lung tissue was harvested from each animal under aseptic conditions; a single-cell suspension was prepared as described above. Single-cell suspensions were stained with antibodies listed below, $Ly6G^+$ viable neutrophils, as well as subpopulations $Ly6G^+CD62L^{high}$, $Ly6g^+CD62L^{low}$ were sorted using a FACS Aria cell sorter (BD Biosciences, BD, New Jersey, U.S.), and the purity of cells was assessed (≥95%). All neutrophils are $CD11b^+$. After sorting, cells were used in a pellet for proteomics or resuspended in DMEM containing 10% FCS for functional assays.

**Isolation of bone marrow neutrophils.** Neutrophils were isolated from the bone marrow of non-infected mice. Bone marrow cells were collected via perfusion of the femoral bones from each animal under aseptic conditions. Cells were meshed through 50 μm filters (Cell Trics, Partec, Sysmex Europe GmbH, Goerlitz, Germany) and erythrocytes lysed in ACK buffer containing $NH_4Cl$ 150 mM, $KHCO_3$ 10 mM, $Na_2EDTA$ 0.1 mM. Single-cell suspensions were stained with the antibodies listed below. $Ly6G^+$ viable neutrophils were sorted using a FACS Aria cell sorter (BD Biosciences, BD, New Jersey, U.S.), and the purity of cells was assessed (≥95%). All neutrophils are $CD11b^+$. After sorting cells were resuspended in DMEM containing 10% FCS.

**Isolation of bone marrow progenitors and in vitro maturation assay (adapted from ref. [16]).** Murine BM progenitor cells were negative selected by depletion of $CD3e^+CD45R^+NK1.1^+CD11b^+Ter119^+$ BM cells using the Streptavidin MicroBeads and LD Columns from Miltenyi Biotec (Bergisch Gladbach, Germany) according to the manufacturer's protocols. For maturation, isolated cells were cultured in 24-well plates at a concentration of $0.3 \times 10^6$ cells/ml in tumor cell line-condensed media (MOPC, GMOPC, or control M-medium) with the addition of mrSCF and mrIL3 (all from Peprotec, Hamburg, Germany; end concentration 50 ng/ml) for 7 days at 37 °C and 5% $CO_2$; the medium was changed into fresh at day 4. aG-CSFR in concentration (5 μg/ml), LLL12 (1 μM), FK866 (100 nM), OT-82 (10 nM), NMN (100 μM) were used. At day 4, tumor-condenced medium was exchanged for sterile M-medium. At day 7, the composition of the cultivated cells in all conditions was analyzed with microscopy and flow cytometry. Phenotype (viability, Ly6G, CD11b, CD62L expression) and functions (ROS, phagocytosis) of $Ly6G^+$ cells were evaluated with flow cytometry as described previously.

For morphological analysis, cytospin preparations of mature bone marrow cells on SuperfrostTM slides (Gibco, Thermo Fisher Scientific, Waltham, MA, US) were fixed with pure methanol, stained with Giemsa (Sigma-Aldrich, Merck KGaA, St. Louis, MO, US), and nuclear morphology was assessed using a light microscope Olympus BX5 (Olympus, Tokyo, Japan). At least 10 fields of view were counted, and percentages of immature, band, and segmented nuclei were calculated.

**Visualization of mitochondria.** Staining for nuclei, cell membrane, and mitochondria with PureBlu Hoechst 33342 Nuclear staining Dye (BioRad), PKH (Sigma-Aldrich/Merck), and Mitospy (BioLegend), respectively, was performed according to the manufacturer's protocols. Samples were evaluated with flow cytometry and on cytospins microscopically.

**Mitochondrial reactive oxygen species.** Cells were washed and resuspended in DMEM containing 10% FCS. *P. aeruginosa* PA14 WT MOI 10 was added. Sterile medium was used as a negative control. ROS production by $Ly6G^+$ viable neutrophils was estimated after 60 min of exposure to *P. aeruginosa* using Dihydrorhodamine 123 (Sigma-Aldrich/Merck, Darmstadt, Germany) with flow cytometry.

**Extracellular reactive oxygen species production measurement by Amplex Red.** Neutrophils were isolated from lung single-cell suspensions as described above and seeded into flat-bottom 96-well plates at a density of $5 \times 10^4$ cells per well in a final volume of 50 μl PBS. *P. aeruginosa* PA14 WT MOI 10 was added, and cells were incubated for 1 h at 37 °C in a humidified atmosphere containing 5% $CO_2$. Amplex Red reagent and horseradish peroxidase (HRP) were then added directly to the wells to achieve the final concentrations recommended by the manufacturer (Thermo Fisher Scientific). ROS-dependent signal was recorded as optical density (OD) at 560 nm using a Synergy2 microplate reader (BioTek Instruments).

**Phagocytosis.** Lung tissue was harvested from non-infected animals under aseptic conditions; a single-cell suspension was prepared and stained with antibodies. Cells were then washed and resuspended in DMEM containing 10% FCS and DNase (to prevent binding of non-phagocytosed bacteria in NETs and false-positive results). Phagocytosis of *P. aeruginosa* PA14 WT labeled with CFSE (MOI 10), or FITC-labeled beads (Caymann, according to manufacturer protocol), or pHrodo™ 647 *E. coli* BioParticles Phagocytosis Kit (Thermo Fisher Scientific, following the manufacturer's protocol) by Ly6G$^+$ neutrophils was estimated after 60 min using flow cytometry.

**NETs release.** Isolated neutrophils 15,000/well were incubated with *P. aeruginosa* (MOI 10) in a glass-bottom 96-well plate (MatTek Corporation, Massachusetts, U.S.) pre-coated with poly-D-lysine 1 mg/ml (Sigma-Aldrich/Merck, Darmstadt, Germany) for 4 h at 37 °C, 5% $CO_2$. Sterile medium was used as a negative control. Samples were fixed with paraformaldehyde (Thermo Fisher Scientific, Massachusetts, U.S.) to a final concentration of 4%, permeabilized with Triton X-100 (Sigma-Aldrich/Merck, Darmstadt, Germany) 0.2% containing buffer. Since the visualization of NETs using DNA-intercalating dyes alone has the risk of detection of necrotic cells or the generation of artificial results based on dye-blocking peptides associated with NETs, antibody-based techniques are required to visualize NETs. Anti-histone 1 antibodies (Merck Millipore, Darmstadt, Germany) were used to detect all NETs. Donkey-anti-mouse-AF564 (Invitrogen, Thermo Fisher Scientific, Massachusetts, U.S.) was used as a secondary antibody. Stainings were mounted with ProLong Gold Antifade Mountant with DAPI (Invitrogen, Thermo Fisher Scientific, Massachusetts, U.S.). Percent of NET-producing cells and NETs length and area were estimated by microscopy, followed by analysis with ImageJ Fiji software.

**Actin branching inhibition with CK-666.** Single-cell lung suspensions were prepared as described above and resuspended in DMEMc 1 mln/ml. CK-666 (MedChemExpress), an Arp2/3 complex inhibitor, was dissolved in DMSO and added to the suspensions at a final concentration of 100 μM to inhibit actin branching, based on established in vitro protocols. Cells were incubated for 24 h at 37 °C in a humidified atmosphere containing 5% $CO_2$. For controls, an equivalent volume of DMSO alone was added to parallel cultures. After treatment, viable Ly6G$^+$ neutrophils were analyzed for phagocytic activity and neutrophil extracellular trap (NET) formation as described above.

**Bacterial killing assessment.** Neutrophils were isolated from lung single-cell suspensions as described above and seeded into flat-bottom 96-well plates at a density of $5 \times 10^4$ cells per well in a final volume of 50 μl PBS. *P. aeruginosa* PA14 WT MOI 10 was added, and cells were incubated for 1 h at 37 °C in a humidified atmosphere containing 5% $CO_2$. Then, the suspension was plated in serial dilutions to estimate CFUs on 2% LB agar and examined after 24 h incubation.

**Bacterial survival in biofilms assessment.** Culture wells were washed with distilled water twice to remove nonattached cells and stained for 15 min by the addition of 0.4% crystal violet (Sigma-Aldrich/Merck,

Darmstadt, Germany) water solution into each well above the initial inoculation, then washed twice with distilled water. Microscopy photographs of dry wells were taken using an AMD EVOS FL digital inverted microscope in brightfield. The area covered with bacteria was calculated with the ImageJ Fiji software. All samples were tested in at least 3 independent wells.

### Proteomic analysis of murine lung samples

**Sample preparation.** One mouse lung lobe each was dissolved in 250 μl of lysis buffer (50 mM Tris (pH 7.8), 150 mM NaCl, and 5% SDS supplemented with complete mini-EDTA-free protease inhibitor, Roche, Penzberg). Samples were sonicated 2 times for 5 min on ice and centrifuged for five minutes at $16,000 \times g$ and room temperature to remove cell debris. A bicinchoninic acid (BCA) assay was performed (Pierce™ BCA Protein Assay Kit, Thermo Fischer Scientific, Waltham, USA) to determine the protein concentration, following the manufacturer's instructions. Tryptic digestion was performed using the single-pot, solid-phase-enhanced sample preparation (SP3) protocol, as described by Hughes et al. (2019).[51] In brief, for a total amount 30 μg protein, disulfide bonds were reduced with a total concentration of 10 mM dithiothreitol (DTT) for 30 min at 56 °C. Free thiol groups at cysteine residues were alkylated with a total concentration of 20 mM iodoacetamide (IAA) for 30 min at 37 °C in the dark. Linear proteins were bound to hydrophilic and hydrophobic beads (50:50 (w/w)); (Sera-Mag™ Carboxylate-Modified Magnetic Beads, Cytiva, Massachusetts, USA) using a bead to protein ratio of 10:1 for 18 min at room temperature. Samples were washed two times with 100% Acetonitrile (LC-MS grade), two times with 70% Ethanol (LC-MS grade), and reconstituted in 50 mM Ammonium Bicarbonate (ABC) buffer in water (LC-MS grade). Tryptic digestion was performed at a Trypsin to protein ratio of 1:100 for 16 h at 37 °C. Peptides were eluted two times with 20 μl 1% FA, 2% Dimethyl sulfoxide (DMSO) in water (LC-MS grade). Peptides were dried in an Eppendorf concentrator plus (Eppendorf SE, Hamburg, Germany) and stored at minus 80 °C until further use. Directly prior to measurement, dried peptides were resolved in 70 μl 0.1% FA and centrifuged at $16,000 \times g$ for 5 min to remove remaining beads. The supernatant was subjected to LC-MS analysis.

**LC-MS/MS data acquisition.** Prior to injection, the peptide concentration was estimated using a NanoDrop Microvolume Spectrometer (Thermo Fischer Scientific, Waltham, USA) at 205 nm. 300 ng of peptides were loaded onto an Evotip (Evosep Biosystems, Odense, Denmark) following the manufacturer's instructions. Briefly, Evotips were rinsed with 0.1% FA in ACN, conditioned with 2-propanol, equilibrated with 0.1% formic acid, and then loaded using centrifugal force at $800 \times g$. Evotips were subsequently washed with 0.1% formic acid, and then 100 μL of 0.1% formic acid was added to each tip to prevent drying. Peptide separation was performed on an Evosep One (Evosep Biosystems, Odense, Denmark) using the predefined Wisper 30SPD method on a PepSep Series column (C18, 15 cm length × 150 μm inner diameter, 1.5 μm particle size (Bruker Daltronics, Billerica, Massachusetts, USA)). Eluting peptides were injected into a TimsTOFpro mass spectrometer (Bruker Daltonics, Billerica, Massachusetts, USA) via a CaptiveSpray source at 1600 V. The mass spectrometer was operated in data-independent acquisition (DIA) PASEF mode. The accumulation and ramp time for the dual TIMS analyzer were set to 100 ms at a ramp rate of 9.42 Hz. Scan ranges were set to m/z 100–1700 for the mass range and $1/k0$ 0.60–1.60 $Vs^{-1}$ $cm^{-2}$ for the mobility range at both MS levels. For DIA-PASEF, a cycle time of 1.8 s was estimated. Within each cycle, a precursor mass range of 400–1200 Da and a mobility range of $1/K0$ 0.6 to 1.6 $Vs^{-1}$ $cm^{-2}$ was fragmented. At the MS1 level, one ramp was executed per cycle. For MS/MS measurements, 32 MS/MS windows with a mass width of 26.0 Da each and a mass overlap of 1.0 were distributed across the 16 MS/MS ramps. LC-MS/MS data processing: LC-MS raw data were processed using the directDIA algorithm in

Spectronaut (Version 19.1.240806.626, Biognosys, Schlieren, Switzerland), applying BSG factory settings (Enzyme/ Cleavage rules: Trypsin/P; Fixed modification: Carbamidomethyl (C); Variable modifications: Acetyl (Protein N-term) and Oxidation(M)), against a reviewed mouse UniProt/Swissprot FASTA database (downloaded on 13th of March 2024, containing 17,196 target sequences). Quantification was performed at the MS2 level. Imputation was disabled. Cross-run normalization was enabled.

**Statistical analysis for LC-MS/MS data.** Normalized protein abundance was used for further downstream analysis, exported, and analyzed in the R software environment (v 4.1.2; R Core Team 2021). Protein abundances were log2-transformed to approach the Gaussian probability distribution. Differentially abundant proteins between multiple predefined phenotypes were determined through analysis of variance (ANOVA). Proteins, identified with a $q$-value (Permutation-based FDR) < 0.05, were considered significant and subjected to further analysis. Differentially abundant proteins between pairs of predefined phenotypes were determined through Student's t-testing. Proteins, identified with a $q$-value (Permutation-based FDR) < 0.05 and at least 1.5-fold change difference between groups, were considered significant and subjected to further analysis.

## Proteomics of isolated lung neutrophils

**Sample preparation.** 80,000 mouse lung neutrophils per sample were dissolved in 70 µl of lysis buffer (50 mM Tris-HCl (pH 7.8)), 150 mM NaCl, and 1% SDS supplemented with complete mini-EDTA-free protease inhibitor, Roche, Penzberg. The proteins were reduced for 30 min at 37 °C in 10 mM DTT and alkylated in 30 mM IAA for 30 min at RT in the dark. After that, the proteins were precipitated with nine volumes of Ethanol for 1 h at −80 °C and centrifuged for 30 min at 20,000 × $g$. The supernatant was removed, and the pellet was dried and dissolved first in 1 µL of 6 M GuHCl and then in 29 µL of 50 mM ammonium bicarbonate buffer, pH 7.8, containing 2 mM CaCl$_2$ and 50 ng of Trypsin (sequencing grade, Promega) and incubated for 18 h at 37 °C. The enzymatic digestion was stopped by acidifying the sample to pH < 2.5 with TFA.

**High pH fractionation.** 8 high pH reversed-phase fractions were created for spectral library generation using the Pierce High pH Reversed-Phase Peptide Fractionation Kit (Thermo Scientific). For fractionation, equal amounts of each analyzed sample were combined to a total of 50 µg of peptides. Peptides were vacuum-dried and dissolved in 0.1% TFA according to the instructions, and fractionation was performed following the manual. Fractionated and vacuum-dried peptide samples were dissolved in 10 µL 0.1% FA for LC-MS/MS measurement.

**LC-MS/MS acquisition.** For liquid chromatography-coupled tandem mass spectrometry (LC-MS/MS) measurements, 2 µL tryptic peptides were injected for individual samples. Spectral library fractions were injected with 2 µL (fraction 1) and 4 µL (fraction 2–8). Measurements were performed on a quadrupole-ion-trap-orbitrap MS (Orbitrap Fusion, Thermo Fisher) coupled to a nano-UPLC (Dionex Ultimate 3000 UPLC system, Thermo Fisher). Chromatographic separation of peptides was achieved with a two-buffer system (buffer A: 0.1% FA in water, buffer B: 0.1% FA in ACN). Attached to the UPLC was a peptide trap (100 µm × 200 mm, 100 Å pore size, 5 µm particle size, C18, Thermo Fisher Scientific) for online desalting and purification, followed by a 25 cm C18 reversed-phase column (75 µm × 250 mm, 130 Å pore size, 1.7 µm particle size, Peptide BEH C18, Waters). Peptides were separated using an 80-min method with linearly increasing ACN concentration from 2% to 30% ACN in 60 min. Eluting peptides were ionized using a nano-electrospray ionization source (nano-ESI) with a spray voltage of 1800, transferred into the MS, and analyzed in data-dependent acquisition (DDA) mode. For each MS1 scan, ions were accumulated for a maximum of 120 milliseconds or until a charge density of $2 \times 10^5$ ions (AGC Target) was reached. Fourier-transformation-based mass analysis of the data from the Orbitrap mass analyzer was performed covering a mass range of 400–1200 m/z with a resolution of 120,000 at m/z = 200. Peptides with charge states between 2+ and 5+ above an intensity threshold of 1000 were isolated within a 1.6 m/z isolation window in Top Speed mode for 3 s from each precursor scan and fragmented with a normalized collision energy of 30% using higher energy collisional dissociation (HCD). MS2 scanning was performed, using an ion trap mass analyzer at a rapid scan rate, covering a mass range starting at m/z 120, and accumulated for 60 ms or to an AGC target of $1 \times 10^5$. Already fragmented peptides were excluded for 30 s.

**Raw data processing and normalization.** LC-MS/MS from DDA were searched with the Sequest algorithm integrated into the Proteome Discoverer software (v 2.4.1.15) (Thermo Fisher Scientific) against a reviewed murine Swissprot database, obtained in October 2020, containing 17053 entries. Carbamidomethylation was set as a fixed modification for cysteine residues, and the oxidation of methionine, and pyro-glutamate formation at glutamine residues at the peptide N-terminus, as well as acetylation of the protein N-terminus were allowed as variable modifications. A maximum number of 2 missing tryptic cleavages was set. Peptides between 6 and 144 amino acids were considered. A strict cutoff (FDR < 0.01) was set for peptide and protein identification. Quantification was performed using the Minora Algorithm, implemented in Proteome Discoverer.

LC-MS/MS from pH fractions were handled in a separate processing step within the software. A multi-consensus workflow was applied to sample and library mgf files to generate a combined output and increase the protein-identification rate for individual samples through feature mapper. Protein abundances for individual samples were exported and submitted to subsequent statistical analysis. Protein abundances for library fractions were discarded prior to normalization. The Lowess algorithm was applied for data normalization.[51]

**Statistics.** Significant regulation was considered for the proteins and transcripts with Log2 FC > ± 2 and $p$-value < 0.05. Subsequently, the $p$-values were adjusted for the false discovery rate (FDR) with Benjamini–Hochberg. The signal-to-noise (SNR) value was calculated using SNR = log2x1-x2d1 + d2, with d1 and d2 representing the respective standard deviations.

Gene ontology (GO) enrichment analysis of the generated datasets of differentially expressed proteins was performed using the open-access ShinyGO platform (Version 0.78, http://bioinformatics. sdstate.edu/go/[52]). The hypergeometric test after the Benjamini–Hochberg false discovery rate (FDR) correction was used to assess statistical significance. Enriched GO terms with FDR-corrected $P$ < 0.05 were considered statistically significant. In addition to the use of functional annotation tools, we also searched PubMed manually to gain insights into the functions of the identified differentially expressed proteins.

Peripheral blood obtained from healthy donors was drawn into 3.8% sodium citrate anticoagulant monovettes (Sarstedt, Nuembrecht, Germany) and mixed 1:1 with PBS (Gibco, Thermo Fisher Scientific, Waltham, MA, US) before separation by density gradient centrifugation (Pancoll density 1077 g/ml). The mononuclear cell fraction was discarded, and neutrophils (purity ≥95%) were isolated by sedimentation over 1% polyvinyl alcohol, followed by hypotonic lysis (0.2% NaCl) of erythrocytes and reconstitution of osmolarity with 1.2% NaCl. Isolated neutrophils (1 mln/ml in RPMI medium containing 10% FCS) alone or with *P. aeruginosa* MOI 10 were incubated for 1 h. Afterward, the analysis of ROS, phagocytosis (as described above), and NET formation (using Sytox Green reagent in combination with anti-MPO according to the manufacturer's protocol) was performed.

Oral rinse with 15 ml of sterile saline was collected from $n = 28$ healthy controls and $n = 45$ patients with HNSCC directly after awaking prior to oral hygiene and food/drink consumption, for 1 min. The absolute number of cells in the rinse was evaluated, and the proportion of viable CD66b$^+$ neutrophils and their activation (CD62L expression) were estimated with flow cytometry. The soluble fraction of the oral rinse was collected and centrifuged at $3000 \times g$. After discarding the supernatant, the bacteria were resuspended in a total of 0.4 ml of sterile saline, and 10 μl of the resuspended bacteria were inoculated onto Columbia blood agar and Chocolate blood agar as universal media, and onto Mac Conkey agar (all media from Oxoid, Wesel, Germany), a selective medium for the growth of Gram-negative rods. Bacteria were cultured at 36 °C under aerobic conditions with 5% $CO_2$, and growth was assessed after 24 and 48 h. Bacteria were identified using VITEK2™ (bioMérieux, Marcy-l'Étoile, France), MicroScan Walk-Away (Beckman Coulter, Brea, US), VITEK MS (bioMérieux) or MALDI Biotyper (Bruker, Billerica, US).

Tumor samples were cut into 1 mm$^3$ pieces with sterile instruments and incubated in RPMI medium containing 10% FCS, 1% Pen/Strep, and 0.2% of Fungizon in proportion 0.02 g/ 0.6 ml for 4 h; afterward, the supernatant was collected. In parallel, a single-cell suspension of the tumor was derived how is explained above; the content and activation (expression of CD62L) of viable tissue CD66b$^+$ neutrophils was evaluated with flow cytometry.Human cohortPatients with head and neck cancer and healthy individuals participated in the study (clinical characteristics in Table 1). No compensation was provided to participants for their involvement in the study.

**ELISA.** G-CSF, MMP9, and TNF-α in murine plasma samples and lung supernatants and neutrophi-conditioned medium, G-CSF in human tumor supernatants and oral rinse were analyzed with ELISA (R&D Systems, Minnesota, U.S.) according to manufacturer protocols.

**RT-qPCR.** The RNA was isolated using Qia Shredder and RNeasy Mini Kit (Qiagen, Hilden, Germany), and the cDNA was produced using the Superscript II Reverse Transcriptase Kit (Invitrogen, Thermo Fisher Scientific, Waltham, MA, US). qRT-PCR was performed at 60 °C annealing temperature using primers listed below. As a housekeeping gene, Rps9 was used. The mRNA expression was measured using the Luna Universal qPCR Master Mix (New England BioLabs, Ipswich, MA, US). Relative gene expressions were calculated by $2^{-\Delta Ct}$ formulations. Real-time RT-PCR was performed using primers listed in the Supplementary Table 2.

List of reagents, antibodies, and consumables in the Supplementary Table 2.

**Analysis of the data deposited in the Gene Expression Omnibus databases.** We used previously published microarray data deposited in the Gene Expression Omnibus databases GSE11247 (GSM283955, GSM283956, GSM283957, GSM283958, GSM283959, GSM283960, GSM283961, GSM283962).

**Statistics.** Statistical analyses were performed using Kruskal–Wallis ANOVA for multiple comparisons with the Bonferroni correction, two-sided Mann–Whitney U-test or $X^2$ test for comparison of quantitative and qualitative parameters, respectively, in independent samples, and Wilcoxon test for dependent samples; correlations were analyzed with Spearman's R test. Sensitivity, specificity of the diagnostic test, and relative risk were calculated using MedCalc's calculators (available online https://www.medcalc.org/calc/). To assess the accuracy of model predictions, receiver operating characteristic (ROC) analysis was performed. Qlucore Omics Explorer 3.10 and Prism 8.0 software were used. $P < 0.05$ was considered significant.

No statistical method was used to predetermine sample size.
No data were excluded from the analyses.

The experiments with human material were not randomized. In murine experiments, mice were randomized into the groups. The investigators were blinded to allocation during experiments.

**Reporting summary**

Further information on research design is available in the Nature Portfolio Reporting Summary linked to this article.

## Data availability
The mass spectrometry proteomics data have been deposited to the ProteomeXchange Consortium via the PRIDE partner repository with the dataset identifiers PXD052631 and PXD069569. Source data are provided with this paper.

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

## Acknowledgements

The study is supported by J.J. grants from the Deutsche Forschungsgemeinschaft (DFG/ JA 2461/2-1, DFG/ JA 2461/7-1, TR332 A5) and Deutsche Krebshilfe (111647). B.S. and H.S. received funding from the INST 337/15-1, INST 337/16-1, INST 152/837-1, and INST 152/947-1 FUGG. M.S. received funding from the TRR332 C2. D.R.E. received funding from the Deutsche Forschungsgemeinschaft FOR5427 SP4; EN984/15-1, 16-1, and 18-1; TR296 P09; TR332 A3 and Z1, and INST 20876/486-1. O.S. received funding from the Deutsche Forschungsgemeinschaft: FOR5427 SP1.

We acknowledge support by the Open Access Publication Fund of the University of Duisburg-Essen, the Imaging Center Essen (Alexandra Brenzel and Dr. Anthony Squire), and the Immunoproteomics group (Stephanie Tautges-Schaefer, Stephanie Thiebes, and Jenny Dick).

## Author contributions

Conceptualization, E.P. and J.J.; Methodology, E.P., O.S., H.H., C.K., M.S., J.K., D.R.E. and J.J.; Software, O.S., B.S., H.S., H.V.; Validation, E.P. and

J.J., Formal Analysis, E.P., L.T., J.R., O.S. and M.S.; Investigation, E.P., L.T., J.R., O.S., N.K., I.T., J.K., I.O., B.S., H.V., H.S. and C.H.; Resources, O.S., B.S., H.S., H.H., S.M., J.K., D.R.E., S.L. and J.J.; Data Curation, E.P., O.S., B.S., H.S. and J.J.; Writing – Original Draft Preparation, E.P., O.S. and C.K.; Writing – Review & Editing, H.H., S.M., M.S., J.K., D.R.E., S.L. and J.J.; Visualization, E.P., O.S., B.S. and H.S.; Supervision, S.L. and J.J., Project Administration, J.J.; Funding Acquisition, O.S., D.R.E., S.L. and J.J.

## Funding

## Competing interests
The authors declare no competing interests.
