## [Transparent Peer Review file · Nature Communications]

G-CSF/NAMPT signaling drives neutrophil dysfunction and enhances bacterial infection susceptibility in cancer patients

Corresponding Author: Professor Jadwiga Jablonska

Version 0:

Reviewer comments:

Reviewer #1

(Remarks to the Author)

The manuscript by Pylaeva et al. entitled "NAMPT/NAD⁺ signaling drives neutrophil dysfunction and enhances bacterial infection susceptibility in cancer patients" reports an interesting investigation showing how elevated G-CSF production by tumors was associated with the persistence of Gram-negative pathogens in head and neck squamous cell carcinoma patients. The presence of Gram-negative pathogens in oral rinse positively correlated with poor prognosis of such patients. Neutrophils are the main immune cells involved in the suppression of bacterial activity, however the authors showed that the activity of this immune population is impaired by tumor cells. This mechanism involved the activation tumor-dependent of the NAMPT/NAD⁺ signaling axis in neutrophils, leading to impaired phagocytosis and NETosis.

The main issue in my opinion is that all the experiments (well conducted with significant results) in the paper focused the attention on G-CSF production and activity and not on NAMPT/NAD axis. The title of the paper does not represent the main finding of the manuscript. The results corresponding to the Figure 1 to 5 are related to the impact of G-CSF on the functionality of neutrophils. The NAMPT/NAD axis is described only in Figure 6, linked with the previous paper published by the same authors (doi: 10.1002/ijc.31808) demonstrating that Nampt/NAD⁺ axis is essential for the activity of G-CSF signaling. In this paper the authors showed a significant accumulation of NAD⁺ in GMOPC bearing mice (Figure 6A), suggesting elevated activity of Nampt. They used NAMPT inhibitor FK866 evaluating the infiltration of neutrophils. Treatment of GMOPC-bearing mice with FK866 decreased lung infiltration with CD62L^{low} neutrophils and improved bacterial clearance (Figure 6K) and clinical performance (Figure 6L), in comparison to untreated mice.

This data revealed that the focus of the paper is G-CSF and not NAMPT/NAD axis.

I suggest to the authors to significantly revised the text of the manuscript, change the title and focusing the abstract and the paper highlighting the main message and finding: the functional role of G-CSF signaling drives neutrophil dysfunction.....

The connection with NAMPT/NAD is very limited. The only use of FK866 is poor. If the authors would like to stress this relationship NAMPT-GCSF they should add several experiments and expand this part of the paper:

1. To repeat all the experiments in mice using a second NAMPT inhibitor, such as OT-82, to confirm the results obtained with FK866
2. To perform a rescue experiment using nicotinamide mononucleotide (NMN) to bypass NAMPT inhibition
3. To measure NAMPT activity, and not only expression
4. The accumulation of NAD, as reported, could not be dependent on NAMPT activity. The authors should measure activities and expression of main NAD-consuming enzymes (PARP/CD38/sirtuins) to evaluate an impairment of NAD consumption rather than biosynthesis. There are several papers showing an impact of CD38 on neutrophils functions. NAD homeostasis is a dynamic balance between synthesis and consumption.
5. It would be very interesting to evaluate progenitors maturation and reprogramming in the presence of NAMPT inhibitors (FK866 and OT-82).
6. To perform an experiment with G-CSF and recombinant NAMPT (high levels in cultured media) to evaluate synergistic or additive effects on neutrophils functions.

In brief, the paper is interesting and technically sound, but I suggest a deeper revision.

Reviewer #2

(Remarks to the Author)

The authors are investigating why cancer patients are more susceptible to bacterial infections. Bacterial infections during

cancer delay and impact treatment outcomes and ultimately mortality and is therefore a hugely important area of research. Moreover, the authors have selected clinically relevant gram-negative bacteria strains, which are in the WHO priority list, which make their study even more exciting. They have formulated an interesting hypothesis that chronic G-CSF secretion by tumor cells is responsible for defected neutrophil responses to bacterial infection and produced some convincing evidence that chronic GCSF has an influence on neutrophil functionality in the lungs of tumor-bearing mice. Thus, the study had a clear purpose and clinical relevance. However, some results lack a clear mechanism which we believe should be addressed by some additional experiments. We also found the paper could be edited to a higher standard which would assist following and interpreting the results.

Major revisions

(1) The figure legends are poorly written, such that it is difficult to follow the paper. The legends should adhere to Nature Communications guidelines 'Figure legends should be <350 words each. They should begin with a brief title sentence for the whole figure and continue with a short statement of what is depicted in the figure, not the results (or data) of the experiment or the methods used. Legends should be detailed enough so that each figure and caption can, as far as possible, be understood in isolation from the main text.'

(2) The premise of the study is that tumors secrete chronically high levels of GCSF and that this alters the function of neutrophils elsewhere e.g., in the lung where they fight bacterial infection. In Fig1D the concentration of GCSF is indeed very high in the HNSCC tumor (how was it collected?), however the blood levels in the HNSCC patients are equal to that of a healthy person. How do the authors explain how the lung neutrophils are altered by high GCSF if it does not enter the blood? Likewise, is the high GCSF found in the oral rinse in 1E coming directly from the tumor into the saliva? The authors discuss how this programming likely happens in the progenitor state (Fig 5) but does GMOPC induce changes in the bone marrow and blood neutrophils? Are GCSF levels increased in bone marrow? How is the differentiation skewed?

(3) In Fig 2H FACS data shows the appearance of an aged CD62Llo CXCR2lo CD11bdim neutrophils. Later the authors postulate possible "dysregulated maturation" based on the proteomic analysis which suggest impaired functional immune and phagocytic responses, which would suggest more immature phenotype of neutrophils. To formally conclude whether their "aged" neutrophils are indeed mature, the expression of CD101, Ly6G and possibly other maturation markers, as well as CXCR4 should also be assessed. Do these neutrophils still look mature (e.g., CD101HI)? Do they start expressing CXCR4? Have you checked their nuclear morphology for segmentation? How is the expression of granule proteins (e.g., from the proteome analysis)?

(4) The authors state that 'GNeutrophils show impaired phagocytosis and NET formations due to defects in cytoskeleton formation' and that high levels of G-CSF results in 'Cytoskeletal-dependent functions being impaired' in lung neutrophils. These statements are based on correlations between the decrease in gene expression of actin-related proteins and F actin with decreased phagocytosis and NET formation. However, there is no evidence to support the two are causally linked. More experiments should be added to prove a direct link and support these claims.

(5) The authors consistently see increased evidence of MPO expression and intracellular ROS activity in Gneutrophils. However, more experimental evidence are required to link this directly to the lung tissue pathology observed. Firstly, the intracellular ROS could be important for killing phagocytosed bacteria (not assessed), while extracellular ROS production and MPO degranulation (released in supernatant) may be more important for tissue damage. Could a bacterial killing assay be conducted? Could the extracellular ROS levels be assessed? In addition, what is the evidence of oxidative damage in the lungs? Could this be assessed by measuring protein or lipid peroxidation?

(6) The authors find upregulation of MMP9 in Gneutrophils, and in the plasma of GMOPC mice. They suggest this also contributes to the tissue damage seen in the lungs. How was MMP9 selected as a gene of interest and is this the only MMP upregulated in neutrophils? Are the matrix proteins remodeled in the lung? Can this be addressed with staining of images of lung – or in vitro culture with a matrix protein important to lung integrity?

(7) The authors show the Gneutrophils are aged, and degranulate more (e.g., Fig 4E) as per decreased side scatter. Can more markers of degranulation be assessed than just SSC, e.g., MPO and NE released in supernatant to directly prove this claim?

(8) A major hypothesis across the paper is that chronic GCSF drives aged tissue damaging neutrophils, that phagocytose less. In Fig 5H culture of GMOPC-neutrophils does not result in impaired phagocytosis, however when the conditioned media is removed for 3 days (in 5M) it does. How do the authors explain this difference? Furthermore, what is the mechanism by which the authors can explain how neutrophils control bacteria when GCSF signaling is inhibited? In Fig6, GCSF inhibition reduces intracellular ROS production and reduces BAL CFUs, and improves clinical score. How do the authors postulate this has happened? Is there evidence of increased biofilms, as suggested in discussion? What mechanism do neutrophils use to kill bacteria? Instead of phagocytosis have the authors performed a bacterial killing assay?

Minor revisions

- Line 758 – ref needs adding
- Table 1 – the table parameters are not explained in the legend e.g., what is UICC?
- Fig 1B in legend should say 1C
- Fig1D – is this digested tumor, or tumor in culture? For blood are the cells cultured or is it the blood plasma levels?
- Fig 2A shows 'GCSF levels in tumor supernatants' what does control refer to, PBS? If so, how is the tumor supernatant

harvested in these samples? Why is the G-CSF conc the same in PBS and cancer? How were the low and high patients selected?

- Fig 2B, why are there no error bars on control and MOPC?
- Fig 2C, D how was the aerated area measured? Please add this to methods.
- Fig 2F likewise the clinical performance needs to be defined in the methods
- Fig 2H = how was this made – is this a TSNE of just neutrophils? Can the statistics be shown?
- Fig 2 A – the G-CSF concentration in tumor supernatants, in this case what sample type is used for the control group?
- Fig 2G legend – does steady state refer to non-infected?
- Fig 2I – what are the blue and orange groups in the PCA?
- Fig 3A – are the blue not upregulated and orange downregulated? How did you select the genes, is this a published dataset?
- Fig3E – why was Histone 1 chosen to assess NETs, when citrullinated H3 is more accurate?
- Fig3F – what are these units and relative to what?
- Fig 3M – legend says this is plasma but text says it is lung homogenates
- Fig 4A – can the line of genes considered significant please be added
- Fig 4K has a very high N, yet the same analysis in Fig 3C has a low N – how are these analyses approached
- Line 395-6 needs a reference
- Fig 5B, can the number of mature segmented neutrophils be quantified out of total. Likewise, in 5C, can the %CD101hi in Ly6G be quantified to prove increased maturation?
- Fig 5F – what is the relevance of measuring CD11b?
- Fig 5H is not significant but line 327-328 says ‘decreased phagocytic activity’
- Fig 6A is not significant but text in line 398 says it is
- Supp 2 B and C why do they have different N number, should they not be the same samples. Can this be explained in legend instead of the finding. The rest of the findings there is not enough information in the legend to properly follow this.
- S3 A – what are the difference between the 2 FACS plots shown? B and C there is not enough detail in the legend to follow these graphs.
- S4 there is simply not enough information to follow, the graphs need labelling, the color scheme explaining and proper legend adding
- S5 add the statistical tests used
- S6A what the units for CD11b? What is ‘N normalized’ data? Line 912 both CD11b and cytometry misspelt.
- S8 more details needed on experimental setup in legend, in B how was the expression measured and normalized? What do the units refer to?
- S9 the lower row, the graphs are overlapping so the legend obscures the graphs. Have any statistical tests been performed? These should be shown.
- S10 lacks information in legend to follow
- S13 what does MTEC and M4 refer to? They are not mentioned in the methods?

Reviewer #3

(Remarks to the Author)

Review of “NAMPT/NAD⁺ signaling drives neutrophil dysfunction and enhances bacterial infection susceptibility in cancer patients”

This manuscript addresses an important and clinically relevant problem: the increased susceptibility of cancer patients to bacterial infections, particularly in the context of head and neck squamous cell carcinoma (HNSCC). The authors convincingly demonstrate that tumor-driven NAMPT/NAD⁺ signaling alters neutrophil function, leading to impaired bacterial clearance, increased tissue damage, and persistence of Gram-negative pathogens. By linking elevated G-CSF production to NAMPT activation and dysfunctional neutrophil phenotypes, the study offers a compelling mechanistic explanation and therapeutic rationale for targeting this axis.

The paper is well written, methodologically sound, and makes a novel contribution to our understanding of tumor-imposed immune dysregulation. The focus on neutrophils, a highly plastic but often overlooked population in cancer immunology, is particularly commendable.

I have only a few minor suggestions to further improve the manuscript:

Potential Confounding by Smoking and Sex in Human Cohorts:

The healthy control group differs notably from the HNSCC group in terms of smoking status and sex distribution (only 35% smokers and 52% male in controls vs. 65% smokers and 78% male in HNSCC). As both factors are known to influence neutrophil function and oral microbiota, these discrepancies could represent confounding variables. I suggest a statistical assessment to evaluate their impact on the observed differences. Ideally, the authors could expand the control cohort to better match for sex and smoking history.

Beyond Antibacterial Functions – Immunomodulatory Roles of Neutrophils in Cancer:

While the study convincingly demonstrates the impact of NAMPT/NAD⁺ signaling on neutrophil antibacterial functions, additional roles of neutrophils in the tumor microenvironment, including antigen presentation, immunosuppression, and modulation of T cell activity are well established. It would enhance the manuscript if the authors could address whether tumor-driven NAMPT activity also skews neutrophils toward pro- or anti-tumoral immune phenotypes, for instance by evaluating expression of markers like MHC class I/II, PD-L1, or Arginase-1. Even a brief discussion of this possibility would

strengthen the paper's broader implications in cancer immunology.

Overall, this is a strong manuscript that uncovers a novel immunometabolic mechanism of neutrophil dysfunction in cancer patients, with clear translational relevance. I recommend acceptance after addressing these minor points.

Version 1:

Reviewer comments:

Reviewer #1

(Remarks to the Author)

The paper is well written and fluent, and the results were significantly improved during this revision. The authors answered to my main questions and suggestions. I'm satisfied with their novel results included in the revised version. The paper is very interesting and is now acceptable for publication.

Reviewer #2

(Remarks to the Author)

The authors did a good job in addressing most of the concerns raised and improving the clarity and impact of the manuscript.

Reviewer #3

(Remarks to the Author)

I thank the authors for the intensive rebuttal. All my concerns have been addressed.

Reviewer #1

All the experiments (well conducted with significant results) in the paper focused the attention on G-CSF production and activity and not on NAMPT/NAD axis. The title of the paper does not represent the main finding of the manuscript.

I suggest to the authors to significantly revise the text of the manuscript, change the title and focusing the abstract and the paper highlighting the main message and finding: the functional role of G-CSF signaling drives neutrophil dysfunction...

Thank you for your thoughtful and constructive feedback regarding the focus of our manuscript and the representation of our main findings in the title and abstract.

We appreciate your suggestion concerning G-CSF, indeed tumor-released G-CSF is a main factor influencing neutrophil dysfunction in cancer. While we also performed the recommended experiments to clarify the role of NAMPT/NAD⁺ signaling and confirmed its involvement as a downstream mediator in this process, we fully agree that the central and novel aspect of our work is the identification of G-CSF as the primary driver of neutrophil reprogramming and impaired antibacterial function. In response to your suggestion, we have taken the following actions; changed as requested the manuscript title: "**G-CSF/NAMPT signaling drives neutrophil dysfunction and enhances bacterial infection susceptibility in cancer patients**", and updated the abstract and main text. We believe these changes more accurately convey the main findings and significance of our work

To repeat all the experiments in mice using a second NAMPT inhibitor, such as OT-82, to confirm the results obtained with FK866.

Thank you for this important suggestion to repeat the key experiments with a second NAMPT inhibitor to confirm the specificity and robustness of our findings obtained with FK866. As suggested, we conducted a comprehensive series of in vivo and in vitro experiments using OT-82, a structurally distinct NAMPT inhibitor. Our results with OT-82 closely mirrored those observed with FK866:

In vitro maturation assays: We performed in vitro maturation assays using OT-82 during neutrophil differentiation from progenitor cells. OT-82 treatment normalized neutrophil differentiation rate, phenotype and function, including restoration of CD62L expression, reduction of spontaneous ROS production, and improved phagocytic capacity, further supporting the role of NAMPT activity in neutrophil reprogramming. These results are now included in the results as a Figure S18a, lines 494-499

In vivo studies: Treatment of tumor-bearing mice with OT-82 (20 mg/kg orally, once daily for three consecutive days followed by four days of rest, repeated for two weeks) resulted in a similar repolarization of neutrophils, characterized by reduced accumulation of dysfunctional, tissue-toxic neutrophil subsets in lungs. These effects were consistent with the phenotypic and functional changes previously reported with FK866. Relatively low dosage was used to reduce toxicity, dose correction should be performed to select the optimal dosage effectively targeting G-CSF/NAMPT-affected granulopoiesis. These results are now included into the Results section lines 545-549, Figure S18c

These findings confirm that the effects of NAMPT inhibition on neutrophil polarization and function are not limited to FK866, but can be reproduced with alternative inhibitors, such as OT-82. This strengthens the conclusion that NAMPT activity is a critical and targetable mediator of tumor-driven neutrophil dysfunction in our model.

To perform a rescue experiment using nicotinamide mononucleotide (NMN) to bypass NAMPT inhibition

Thank you for your suggestion to perform a rescue experiment using nicotinamide mononucleotide (NMN) to determine whether the effects of NAMPT inhibition on neutrophil maturation and function can be bypassed by direct supplementation of the NAD⁺ pathway.

In response, we conducted in vitro maturation assays using bone marrow progenitors in the presence of the NAMPT inhibitor FK866, with and without the addition of NMN. Our results demonstrated that NMN supplementation effectively bypassed the effects of NAMPT inhibition, specifically by accumulation of aged CD62L^{low} CD11^{low} phenotype, with elevated production of ROS.

These findings confirm that the observed effects of FK866 are specifically due to disruption of the NAD⁺ biosynthetic pathway and can be reversed by providing NMN as a downstream metabolite. This further supports the mechanistic link between NAMPT activity, NAD⁺ availability, and neutrophil functional reprogramming. These results are now included as a Figure S18b, lines 529-537.

To measure NAMPT activity, and not only expression

In our study, we strongly focused on the functional consequences of NAMPT activity by employing two structurally distinct NAMPT inhibitors, FK866 and OT-82, both in vitro and in vivo. The consistent restoration of neutrophil function and phenotype following NAMPT inhibition strongly supports the notion that elevated NAMPT activity plays a critical role in neutrophil dysfunction within the context of tumor-derived G-CSF exposure. These pharmacological interventions directly target NAMPT enzymatic activity, and their efficacy in reversing neutrophil defects provides functional evidence for NAMPT's involvement beyond expression levels alone.

The accumulation of NAD, as reported, could not be dependent on NAMPT activity. The authors should measure activities and expression of main NAD-consuming enzymes (PARP/CD38/sirtuins) to evaluate an impairment of NAD consumption rather than biosynthesis.

Thank you for your thoughtful comment regarding the potential contribution of impaired NAD consumption, rather than increased biosynthesis, to the observed NAD accumulation. We appreciate your suggestion to assess the activities and expression of key NAD-consuming enzymes, including PARPs, CD38, and sirtuins.

As NAD⁺ accumulation can be due to decreased NAD⁺ consumption, we analyzed the abundance of major NAD-consuming enzymes in the whole lung proteome. The PARP/CD38/sirtuins pathway was not significantly altered in the lung tissue (Figure S18c). The efficacy of inhibition of NAMPT activity in reversing neutrophil defects provides functional evidence for NAMPT's involvement.

It would be very interesting to evaluate progenitors maturation and reprogramming in the presence of NAMPT inhibitors (FK866 and OT-82).

Thank you for your suggestion to investigate the effects of NAMPT inhibitors on progenitor maturation and reprogramming. In the manuscript, we evaluated the impact of FK866 on neutrophil progenitor maturation and functional reprogramming, as detailed in the relevant figures (Figure 6b). To further address this point and confirm the specificity of our findings, we also performed additional experiments using OT-82, a structurally distinct NAMPT inhibitor.

Our results demonstrate that OT-82 influences progenitor maturation and neutrophil reprogramming similarly to FK866. Both inhibitors normalized neutrophil phenotype (CD62L expression) and function (ROS and, in case of OT82, also phagocytosis) during in vitro maturation, supporting the conclusion that NAMPT activity is a critical mediator of the observed changes. These results are now included into FigureS18a, lines 494-499.

These findings confirm the robustness and reproducibility of our results, and further strengthen the mechanistic link between NAMPT signaling and neutrophil dysfunction in the context of tumor-derived G-CSF exposure.

To perform an experiment with G-CSF and recombinant NAMPT (high levels in cultured media) to evaluate synergistic or additive effects on neutrophil functions.

Thank you for your suggestion to evaluate the potential synergistic or additive effects of G-CSF and recombinant NAMPT on neutrophil functions by performing experiments with both factors at high concentrations in cultured media.

We carefully considered this experimental approach. However, extracellular NAMPT is known to act as a ligand for Toll-like receptor 4 (TLR4) [10.1038/s41467-019-12055-2], independently of its enzymatic activity in the NAD⁺ biosynthetic pathway. This property introduces an additional immunomodulatory effects that are distinct from the intracellular NAMPT/NAD⁺ axis explored in our study. Including recombinant NAMPT at high levels in the culture medium would therefore increase the complexity of the experimental system, making it challenging to interpret the specific contributions of G-CSF-driven NAMPT signaling versus the direct, TLR4-mediated effects of extracellular NAMPT.

To maintain a clear mechanistic focus and avoid confounding factors, we decided not to pursue this combined treatment experiment. We appreciate your understanding and the opportunity to explain our decision-making process regarding experimental design.

Reviewer #2

Some results lack a clear mechanism which we believe should be addressed by some additional experiments

Thank you for your comment regarding the need for clearer mechanistic explanations underlying some of our results. We appreciate your emphasis on the importance of mechanistic validation to strengthen the overall conclusions of the study.

In response, we have performed additional experiments specifically designed to address the mechanistic gaps identified in our initial findings. These experiments were aimed at elucidating the pathways and molecular interactions responsible for the observed effects, thereby providing a more comprehensive understanding of the underlying biology. We have integrated these new results into the revised manuscript, with updated figures and expanded discussion to clearly present the mechanistic insights gained. We believe these additions address your concerns and significantly enhance the scientific rigor and interpretability of our work.

In Fig1D the concentration of G-CSF is indeed very high in the HNSCC tumor (how was it collected?), however the blood levels in the HNSCC patients are equal to that of a healthy person. How do the authors explain how the lung neutrophils are altered by high G-CSF if it does not enter the blood?

Thank you for your insightful question regarding the apparent discrepancy between the high G-CSF concentrations observed in HNSCC tumor tissue (Fig. 1D) and the comparable plasma G-CSF levels in HNSCC patients and healthy controls. We appreciate the opportunity to clarify both the methodology and our interpretation of these findings.

The high G-CSF concentrations reported for tumor tissue were determined by collecting tumor-conditioned medium. As described in the Methods section, freshly resected HNSCC tumors were mechanically sectioned into 1 mm³ cubes and incubated in culture medium (with the volume normalized to tumor mass) for 4 hours. The supernatant was then collected and analyzed for G-CSF content (ELISA). This approach allows for the assessment of G-CSF produced and released by the tumor microenvironment under ex vivo conditions, reflecting the local cytokine milieu to which infiltrating immune cells are exposed.

Although tumor tissue releases substantial amounts of G-CSF, our measurements did not reveal significant differences in plasma G-CSF concentrations between healthy individuals and HNSCC patients. This may be explained, at least in part, by the detection limit of the ELISA method used (20 pg/mL), with measured values around 10 pg/mL in both groups—below the assay's sensitivity threshold. Thus, subtle or transient increases in circulating G-CSF, or compartmentalized elevations, may not be reliably detected with this standard assay.

Despite the lack of measurable systemic G-CSF elevation, several lines of evidence support the notion that neutrophils in HNSCC patients are exposed to and reprogrammed by tumor-derived G-CSF.

Neutrophils from HNSCC patients in comparison to neutrophils from healthy individuals [GSE79404, 10.1126/sciimmunol.aaf8943] display molecular signatures of active G-CSF signaling (JAK-STAT signaling pathway hsa04630), indicating functional exposure to G-CSF.

Previous studies, including our own work [10.1002/ijc.31808], have documented heterogeneity in plasma G-CSF levels among HNSCC patients, particularly when stratified by disease stage. This suggests that systemic G-CSF may be elevated in subsets of patients or at specific disease stages, which could contribute to neutrophil reprogramming in those contexts.

The tumor microenvironment can create high local concentrations of G-CSF, particularly within the tumor and draining tissues. Neutrophil progenitors [10.1126/science.adf6493] trafficking through or residing in these areas can be directly influenced by these elevated cytokine levels, even if systemic levels remain low.

Another aspect of this is that G-CSF may reach the bone marrow or peripheral tissues not only as a soluble factor in plasma but also encapsulated within extracellular vesicles (EVs). There is direct evidence that extracellular vesicles (EVs) can accumulate in bone marrow following their release from distant tissues (10.1002/jev2.12223). This accumulation in BM during inflammation is described as notable and significant, beyond mere passive distribution in plasma, with enhanced functional consequences for hematopoietic and immune cell dynamics. Such vesicle-mediated transport can facilitate targeted cytokine delivery and may evade detection by conventional ELISA assays. Our unpublished data support the presence of G-CSF within EVs in HNSCC, which could explain the observed neutrophil phenotypes and at the same time the limitation to detect it in the blood using our detection method.

We have discussed these points in the revised manuscript (lines 585-602) and appreciate your attention to this important aspect of our study.

Likewise, is the high G-CSF found in the oral rinse in 1E coming directly from the tumor into the saliva?

Thank you for this important question. Indeed, our data indicate that the high concentration of G-CSF found in the oral rinse of HNSCC patients (varying between 100-1000 pg/ml) is most likely derived directly from the tumor microenvironment rather than from systemic circulation (where the levels are around 10 pg/ml) (lines 581-584).

The authors discuss how this programming likely happens in the progenitor state (Fig 5) but does GMOPC induce changes in the bone marrow and blood neutrophils?

Thank you for your question regarding whether GMOPC induces changes in bone marrow and blood neutrophils, in addition to the programming observed at the progenitor stage (as shown in Figure 5).

We addressed this point by directly analyzing neutrophils from both the bone marrow and peripheral blood of GMOPC- and MOPC-bearing mice. Our investigations revealed that bone marrow and blood neutrophils from GMOPC-bearing mice exhibited changes highly similar to those observed in lung neutrophils and in the in vitro maturation system. There was a marked increase in the proportion of aged, CD62L^{low} CD11b^{low} neutrophils in both bone marrow and blood of GMOPC-bearing mice, mirroring the phenotype seen in lung tissue. The observed alterations in neutrophil phenotype were present in all analyzed compartments—bone marrow, blood, and lung—indicating that the programming initiated by tumor-derived G-CSF is systemic and not restricted to a single tissue. We included these results now in the Figure S4, lines 346-350.

These findings support the conclusion that chronic exposure to tumor-derived G-CSF (as modeled by GMOPC) induces persistent and widespread reprogramming of neutrophils, beginning at the progenitor level in the bone marrow and manifesting in both circulating and tissue-resident neutrophil populations.

Are G-CSF levels increased in bone marrow? How is the differentiation skewed?

Yes, G-CSF levels are increased in the bone marrow (BM) in the context of ^GMOPC-bearing mice, (Figure S4a). Chronic elevation of G-CSF in the BM leads to significant alterations in myeloid differentiation. There is accelerated differentiation of hematopoietic progenitors toward the neutrophil lineage, resulting in an increased output of neutrophils from the BM. The newly generated neutrophils display an “aged” or “exhausted” phenotype.

To formally conclude whether their “aged” neutrophils are indeed mature, the expression of CD101, Ly6G and possibly other maturation markers, as well as CXCR4 should also be assessed. Do these neutrophils still look mature (e.g., CD101^{HI})? Do they start expressing CXCR4? Have you checked their nuclear morphology for segmentation?

We thank the reviewer for this important question regarding the maturation status of ^Gneutrophils. We have carefully assessed classical markers of neutrophil maturation and nuclear morphology in our model.

Representative fluorescence images of neutrophils from ^GMOPC lungs, reveal clearly segmented nuclei (Hoechst staining), which is characteristic of mature neutrophils.

Quantitative flow cytometry analyses demonstrate that neutrophils from ^GMOPC-bearing mice express significantly higher levels of Ly6G and CD101 compared to controls, with the most prominent increases observed in bone marrow, followed by lung and blood compartments. Specifically, ^Gneutrophils display a CD101^{high} phenotype consistent with terminal maturation. ^Gneutrophils also upregulate CXCR4 across all compartments—again, most strongly in bone marrow. This is in line with the profile of “aged” or terminally mature neutrophils, which recirculate or are poised for clearance.

In summary, our data collectively demonstrate that ^Gneutrophils retain a mature phenotype, defined by segmented nuclei and high expression of Ly6G and CD101. These cells also upregulate CXCR4, particularly in the bone marrow. This multi-parameter analysis strongly supports their classification as mature or “aged” neutrophils.

We hope these results address the reviewer’s concern. We included these results in the Results section lines 344-345 and Figure S11.

a ^gMOPC lung neutrophils

c

b

d

How is the expression of granule proteins (e.g., from the proteome analysis)?

The following neutrophil-derived proteins are upregulated in the lungs of G^oMOPC-bearing mice:

Primary (azurophilic) granules-associated proteins detected in G^oMOPC lungs: neutrophil elastase (ELANE), cathepsin G (CTSG), proteinase 3 (PRTN3), and myeloperoxidase (MPO), cathelicidin antimicrobial peptide (CAMP).

Secondary (specific) granules-associated proteins: lipocalin 2 (LCN2, also known as NGAL), lactoferrin (LTF), olfactomedin 4 (OLFM4).

Tertiary (gelatinase) granules-associated proteins: matrix metalloproteinase 9 (MMP9), neutrophil collagenase (MMP8), and matrix metalloproteinase 25 (MMP25 or MT6-MMP).

Secretory vesicles-associated proteins: FCGR3, complement component 5a receptor 1 (C5AR1).

We added these results into the Results section lines 390-392, Figure S12a and Figure S9a

The authors state that 'GNeutrophils show impaired phagocytosis and NET formations due to defects in cytoskeleton formation' and that high levels of G-CSF results in 'Cytoskeletal-dependent functions being impaired' in lung neutrophils. These statements are based on correlations between the decrease in gene expression of actin-related proteins and F actin with decreased phagocytosis and NET formation. However, there is no evidence to support the two are causally linked. More experiments should be added to prove a direct link and support these claims.

Thank you for your important comment regarding the need to establish a direct, causal relationship between cytoskeletal defects and the impaired phagocytosis and NET formation observed in ⁶Neutrophils exposed to high levels of G-CSF.

To address this point, we performed additional functional experiments using CK666, a well-characterized inhibitor of the Arp2/3 complex that specifically blocks actin cytoskeletal branching. Treatment of neutrophils with CK666 resulted in a significant suppression of phagocytosis, directly confirming that proper actin cytoskeleton reorganization is essential for neutrophil phagocytic function. These results provide experimental evidence that defects in actin polymerization and cytoskeletal dynamics, as observed in ⁶Neutrophils, are causally linked to their impaired antibacterial activity.

We have included these new data in the revised manuscript (Figure S7b, lines 249-254) to strengthen our mechanistic claims regarding the role of cytoskeletal reorganization in neutrophil dysfunction under chronic G-CSF exposure. Thank you for prompting us to provide this functional link between cytoskeleton and functions of neutrophils.

The authors consistently see increased evidence of MPO expression and intracellular ROS activity in ⁶Neutrophils. However, more experimental evidence are required to link this directly to the lung tissue pathology observed.

Firstly, the intracellular ROS could be important for killing phagocytosed bacteria (not assessed), while extracellular ROS production and MPO degranulation (released in supernatant) may be more important for tissue damage. Could a bacterial killing assay be conducted? Could the extracellular ROS levels be assessed? In addition, what is the evidence of oxidative damage in the lungs? Could this be assessed by measuring protein or lipid peroxidation?

Thank you for your thoughtful comments regarding the mechanistic link between increased MPO expression, ROS activity in ⁶Neutrophils, and the observed lung tissue pathology. We appreciate your suggestions for further experimental validation and are pleased to address each of your points below.

To directly evaluate extracellular ROS release, we performed the Amplex Red assay, which specifically detects hydrogen peroxide in the extracellular compartment. Our results confirmed that ⁶Neutrophils exhibit significantly elevated spontaneous ROS release compared to control neutrophils. Notably, while spontaneous ROS production and release were higher in ⁶Neutrophils, particularly in CD62L^{low} cells, the increase in ROS in response to bacterial stimulation was less pronounced than in normal

neutrophils (Figure S8c,d, lines 309-315). This suggests a dysregulated oxidative burst, with higher baseline tissue exposure to ROS but impaired antibacterial response upon challenge.

We conducted bacterial killing assays to assess the functional consequences of altered ROS dynamics. Short-term killing (1-hour plating, M4) was more effective in GNeutrophils, indicating that initial antibacterial mechanisms remain active. However, with prolonged exposure (24-hour MTT assay), there was increased evidence of biofilm formation by bacteria (scale bar 100 μ m), suggesting that G^oNeutrophils may fail to sustain effective bacterial clearance over time possibly due to their functional exhaustion (Figure S10).

Our analyses revealed an elevated expression of tissue-damaging molecules in the lungs of G^oMOPC-bearing mice, including increased MMP9 and MPO levels. In parallel, we observed decreased expression of occludin - protein critical for lung tissue integrity, further supporting the presence of tissue injury in this model. The proteomic signature (supplemental file 3) and functional data strongly indicate increased oxidative stress and compromised tissue structure.

We included these results into Figure S9

The authors find upregulation of MMP9 in Gneutrophils, and in the plasma of GMOPC mice. They suggest this also contributes to the tissue damage seen in the lungs. How was MMP9 selected as a gene of interest and is this the only MMP upregulated in neutrophils?

Are the matrix proteins remodeled in the lung? Can this be addressed with staining of images of lung – or in vitro culture with a matrix protein important to lung integrity?

Thank you for your thoughtful question regarding the selection of MMP9 as a gene of interest, the upregulation of other MMPs in neutrophils, and the evidence for matrix remodeling in the lungs.

Our proteomic analysis of lung tissue from GMOPC-bearing mice identified upregulation of MMP8, MMP9 and MMP25 in lung tissue (Figure S9a, lines 317-321).

These MMPs are implicated in ECM remodeling and immune regulation. The concurrent upregulation of these three MMPs suggests a broader pattern of matrix-degrading enzyme induction in the context of chronic G-CSF exposure. MMP9 was chosen as a target molecule based on its established role in the G-CSF/NAMPT signaling pathway, which has been previously published as relevant to neutrophil-mediated matrix degradation and angiogenesis (10.1002/ijc.31808).

Morphological assessment of lung tissue from GMOPC-bearing mice revealed several classic features of matrix remodeling, including disrupted distribution of collagen in the lung tissue, thickening of

alveolar walls, increased cellularity within the lung parenchyma. The left side displays lung tissue with a well-organized, continuous network of collagen fibers. The collagen staining is robust and outlines the alveolar walls and septa clearly, reflecting preserved alveolar architecture and structural integrity. These features are characteristic of normal lung tissue. The right side shows lung tissue with fragmented, patchy, or reduced collagen staining. The collagen network appears disrupted, with weak or absent red staining areas. The alveolar structure is less supported, indicating significant collagen degradation. These changes are typical of MMP9-degraded lung tissue, where matrix metalloproteinase activity has broken down the collagen framework (Figure S9b, lines 325-330).

These features are consistent with active matrix remodeling and tissue injury, as described in the literature [10.3390/cells8040342]. High levels of MMP9 are associated with increased susceptibility to bacterial colonization and persistence in the lung, as demonstrated in both animal models and human studies [10.4049/jimmunol.178.2.1013, 10.1101/2025.04.15.25325903].

The authors show the Gneutrophils are aged, and degranulate more (e.g., Fig 4E) as per decreased side scatter. Can more markers of degranulation be assessed than just SSC, e.g., MPO and NE released in supernatant to directly prove this claim?

Thank you for your valuable suggestion to assess additional, more direct markers of neutrophil degranulation beyond side scatter (SSC). In our study, we quantified MMP9 levels in the supernatant of Gneutrophils, which is a well-established component of tertiary (gelatinase) granules and a marker of neutrophil degranulation. We observed significantly elevated MMP9 release from Gneutrophils, supporting the conclusion that these cells are more prone to degranulation. These data are now included into Figure 4F,G.

A major hypothesis across the paper is that chronic G-CSF drives aged tissue damaging neutrophils, that phagocytose less. In Fig 5H culture of GMOPC-neutrophils does not result in impaired phagocytosis, however when the conditioned media is removed for 3 days (in 5M) it does. How do the authors explain this difference?

Thank you for your careful reading and for highlighting the apparent discrepancy regarding neutrophil phagocytosis in Figures 5H and 5M. We appreciate the opportunity to clarify this point.

In Figure 5H, we observed a trend toward reduced phagocytosis in GMOPC-neutrophils cultured continuously in tumor-conditioned medium. To carefully prove reviewer comment, we repeated our experiment with significantly more samples. Moreover, we aimed to study only effective phagocytosis, for this we used pHrodo™ kit that specifically indicates successful phagosome formation and acidification, a hallmark of productive phagocytosis. It improved the accuracy and demonstrated that in G-MOPC effective phagocytosis is indeed depleted. All in all, the general phenomenon of impaired phagocytosis in neutrophils exposed to chronic G-CSF is reproducible and supported by both in vitro and in vivo data.. We accordingly expanded our results – Figure 5h.

Furthermore, what is the mechanism by which the authors can explain how neutrophils control bacteria when G-CSF signaling is inhibited? In Fig6, G-CSF inhibition reduces intracellular ROS production and reduces BAL CFUs, and improves clinical score. How do the authors postulate this has happened? Is there evidence of increased biofilms, as suggested in discussion? What mechanism do neutrophils use to kill bacteria? Instead of phagocytosis have the authors performed a bacterial killing assay?

Thank you for your insightful questions regarding the mechanisms by which neutrophils regain antibacterial function when G-CSF signaling is inhibited, as shown in Figure 6. We think that the inhibition of chronic G-CSF/NAMPT signaling restores neutrophil antibacterial capacity primarily by reducing the accumulation of dysfunctional, tissue-damaging neutrophils, normalizing spontaneous ROS production, and supporting a more balanced, functionally competent neutrophil pool. This leads to improved bacterial clearance in the lungs, as evidenced by reduced bacterial loads in bronchoalveolar lavage and better clinical scores. Our data indicate that, while short-term bacterial killing by neutrophils may be preserved even in the dysfunctional state, chronic G-CSF exposure creates an environment permissive for biofilm formation and bacterial persistence. These data are now included into the Results section Figure S10, lines 333-338.

Minor comments

We are thankful for the careful revision, all the comments are addressed and corrections and clarifications are included in the manuscript

- Line 758 – ref needs adding / added
- Table 1 – the table parameters are not explained in the legend e.g., what is UICC? / done
- Fig 1B in legend should say 1C / done
- Fig1D – is this digested tumor, or tumor in culture? For blood are the cells cultured or is it the blood plasma levels? / done
- Fig 2A shows ‘GCSF levels in tumor supernatants’ what does control refer to, PBS? If so, how is the tumor supernatant harvested in these samples? Why is the GCSF conc the same in PBS and cancer? How were the low and high patients selected? / done
- Fig 2B, why are there no error bars on control and MOPC? / all the animals cleared the bacteria successfully, so the amount of bacteria in BAL was equally undetectable, and error bars could not be applied.
- Fig 2C, D how was the aerated area measured? Please add this to methods. / added
- Fig 2F likewise the clinical performance needs to be defined in the methods / added
- Fig 2H = how was this made – is this a TSNE of just neutrophils? Can the statistics be shown? / added
- Fig 2 A – the GCSF concentration in tumor supernatants, in this case what sample type is used for the control group? / done
- Fig 2G legend – does steady state refer to non-infected? / yes, corrected
- Fig 2I – what are the blue and orange groups in the PCA? / done
- Fig 3A – are the blue not upregulated and orange downregulated? How did you select the genes, is this a published dataset? / The colorcode is correct, orange color refers to upregulated proteins in the GMOPC neutrophils, blue is downregulated. They are published datasets, please find the lists with description of the datasets in the supplementary files 1-7.
- Fig3E – why was Histone 1 chosen to assess NETs, when citrullinated H3 is more accurate? / After careful consideration, we avoided to use citrullinated histones as a marker of PAD4-dependent NETosis, as it limits the detection of NETs released through alternative mechanisms (de Buhr N, von Köckritz-Blickwede M, 2016).

- Fig3F – what are these units and relative to what? / The abundances of proteins measured by quantitative proteomics are relative to the abundance of the protein in other samples, what is common for the type of proteomic analysis that was performed
- Fig 3M – legend says this is plasma but text says it is lung homogenates /corrected
- Fig 4A – can the line of genes considered significant please be added / the regulated proteins were selected according to Signal-to-noise ratio (SNR) >1 (lines 1039-1040)
- Fig 4K has a very high N, yet the same analysis in Fig 3C has a low N – how are these analyses approached / in Fig 3c the median length of NETs in GMOPC total cells is around 20 um, what is comparable to the values in Fig. 4K
- Line 395-6 needs a reference / added
- Fig 5B, can the number of mature segmented neutrophils be quantified out of total. Likewise, in 5C, can the %CD101hi in Ly6G be quantified to prove increased maturation? / We quantified % of segmented cells after 6 days of in vitro maturation, the trend repeats the findings of % Ly6G+ cells by flow cytometry, the difference could be explained by different experimental settings (microscopy – counting cells with segmented nuclei from total cells, flow cytometry – Ly6G+ from total alive cells). CD101 expression

- Fig 5F – what is the relevance of measuring CD11b? / MFI, corrected
- Fig 5H is not significant but line 327-328 says ‘decreased phagocytic activity’ / corrected
- Fig 6A is not significant but text in line 398 says it is / corrected
- Supp 2 B and C why do they have different N number, should they not be the same samples. Can this be explained in legend instead of the finding. The rest of the findings there is not enough information in the legend to properly follow this. / corrected
- S3 A – what are the difference between the 2 FACS plots shown? B and C there is not enough detail in the legend to follow these graphs. / corrected
- S4 there is simply not enough information to follow, the graphs need labelling, the color scheme explaining and proper legend adding / corrected
- S5 add the statistical tests used / corrected

- S6A what the units for CD11b? What is 'N normalized' data? Line 912 both CD11b and cytometry misspelt. / corrected
- S8 more details needed on experimental setup in legend, in B how was the expression measured and normalized? What do the units refer to? / corrected
- S9 the lower row, the graphs are overlapping so the legend obscures the graphs. Have any statistical tests been performed? These should be shown. / corrected
- S10 lacks information in legend to follow / corrected
- S13 what does MTEC and M4 refer to? They are not mentioned in the methods? / corrected

Reviewer #3

Potential Confounding by Smoking and Sex in Human Cohorts: The healthy control group differs notably from the HNSCC group in terms of smoking status and sex distribution (only 35% smokers and 52% male in controls vs. 65% smokers and 78% male in HNSCC). As both factors are known to influence neutrophil function and oral microbiota, these discrepancies could represent confounding variables. I suggest a statistical assessment to evaluate their impact on the observed differences. Ideally, the authors could expand the control cohort to better match for sex and smoking history.

Thank you for highlighting the potential confounding effects of smoking status and sex distribution between the healthy control and HNSCC groups. We agree that both factors are known to influence neutrophil function and the oral microbiota. To address this concern, we performed a statistical assessment of these variables in relation to the presence of Gram-negative pathogens in the oral cavity among HNSCC patients.

We analyzed the following clinical parameters for their association with the presence of Gram-negative pathogens in the oral cavity within the HNSCC cohort:

Parameter	No Gram-negative Pathogens	Yes Gram-negative Pathogens	p-value
Mean Age (years)	65	66.3	0.87
Male Sex (%)	87.50%	83.30%	0.73
Current Smokers (%)	73.70%	50.00%	0.2
Mean Pack Years	43.3	29.9	0.25
HPV Positive (%)	44.40%	44.40%	1

Mann-Whitney U-Test and X^2 tests were used for comparison of two independent samples. None of these parameters—including sex and current smoking status—were significantly associated with the presence of Gram-negative pathogens in the oral cavity (all $p > 0.05$). These results are included in the Results section, table 2, lines 89-92

Beyond Antibacterial Functions – Immunomodulatory Roles of Neutrophils in Cancer: While the study convincingly demonstrates the impact of NAMPT/NAD⁺ signaling on neutrophil antibacterial functions, additional roles of neutrophils in the tumor microenvironment, including antigen presentation, immunosuppression, and modulation of T cell activity are well established. It would enhance the manuscript if the authors could address whether tumor-driven NAMPT activity also skews neutrophils toward pro- or anti-tumoral immune phenotypes, for instance by evaluating expression of markers like MHC class I/II, PD-L1, or Arginase-1. Even a brief discussion of this possibility would strengthen the paper’s broader implications in cancer immunology.

Thank you for your insightful comment regarding the broader immunomodulatory functions of neutrophils in the tumor microenvironment, beyond their antibacterial activity. We appreciate the opportunity to clarify and expand on this important aspect.

Following Reviewer suggestion, we evaluated the expression of key immunomodulatory markers—including MHC class I, MHC class II, CD80, and PD-L1—on tumor-infiltrating myeloid cells, including neutrophils, in both MOPC and G-MOPC tumor models. Our analyses did not reveal any statistically significant differences in the expression of these markers between the two groups of mice. In majority of markers we don’t see significant alterations. Even though the PDL1 seems to be downregulated in G-MOPC tumor, it could not be responsible for the observed elevated susceptibility to bacterial infection in G-MOPC-bearing mice.

Moreover, in the proteome of peripheral (lung) neutrophils associated with MOPC vs G-MOPC tumors, no immunoactivation or immunosuppression-associated proteins in the list showed statistically significant changes ($p < 0.1$) with either positive or negative fold change in MOPC vs G-MOPC groups (lines 402-404, the Figure S13, supplementary file 1). This suggests that, within the context of our experimental system, chronic tumor-derived G-CSF and associated NAMPT/NAD⁺ signaling do not significantly alter the antigen-presenting or immunosuppressive phenotype of neutrophils and lung tissue as defined by these surface markers.

As we previously demonstrated, NAMPT signaling significantly potentiates angiogenic properties of neutrophils (with elevated expression of VEGF and MMP9), driving their tumor-supporting properties [10.1002/ijc.31808]. The question of G-CSF/NAMPT axis in neutrophil-mediated immunosuppression requires further investigation.

The data are included in the result and discussion sections.